# BEYOND PAIRWISE MODELING: TOWARDS EFFICIENT AND ROBUST TRAJECTORY SIMILARITY COMPUTATION VIA REPRESENTATION LEARNING

## ABSTRACT

Accurate trajectory similarity computation is crucial in ride-sharing applications, where trajectories of varying lengths need to be aligned into a uniform representation. Existing methods suffer from reliance on multi-metric supervision and the role-specific encoding required for triplet loss computation, resulting in inefficient computation. To overcome these issues, we move beyond pairwise modeling and propose a novel representation learning framework to achieve efficient and robust trajectory similarity computation, named Hyper2Edge. Hyper2Edge consists of three main components: (i) Hypergraph-based modeling to represent trajectories as hyperedges, instead of single nodes, preserving sequential and structural details; (ii) Hierarchical trajectory representation learning to capture intra- and inter-trajectory patterns; and (iii) A weighted top-$k$ InfoNCE loss to focus on nearest-neighbor relations, addressing the inefficiencies of triplet loss. Evaluated on two public benchmarks, Hyper2Edge achieves an average absolute gain of 7.42% across all evaluation metrics and an average improvement of 45.9% in accuracy compared to state-of-the-art methods, while maintaining competitive training time per epoch on par with the best-performing methods. The code is available at: https://anonymous.4open.science/r/Hyper2Edge-3D2B.

## 1 INTRODUCTION

In ride-sharing applications, trajectory similarity is commonly computed using representative points rather than the full GPS sequence. However, GPS trajectories are inherently heterogeneous in terms of length and sampling frequency, and extracting representative points may further amplify these variations. Recently, Trajectory Representation Learning (TRL) Jiang et al. (2023); Ma et al. (2024); Zhou et al. (2025b;c) has emerged as a versatile and powerful preprocessing technique. TRL represents trajectories of varying lengths as vectors in a unified dimension, achieving trajectory alignment.

Trajectory similarity computation emerges as a key downstream task of TRL that faces two practical challenges, as illustrated in Figure 1. First, many existing methods Yao et al. (2022); Chang et al. (2023); Chuang et al. (2024); Li et al. (2025) depend on supervised training using specific distance metrics such as DTW Rakthanmanon et al. (2012) or ERP Chen & Ng (2004). In practice, however, trajectory similarity is typically computed uniformly in Euclidean space, where all distance labels are also derived. Using multiple distance metrics as training labels, rather than directly using Euclidean distance, introduces unnecessary computational redundancy. Second, even with a suitable metric, the encoding process itself remains inefficient. Although some approaches Yao et al. (2019); Zhang et al. (2020); Yang et al. (2022) achieve linear time complexity, they must encode each trajectory alongside both positive and negative samples to compute triplet loss. When these samples act as anchor trajectories, they require additional encoding, resulting in repetitive encoding. As a consequence, theoretically efficient methods are often unable to achieve their full potential in practical scenarios.

To address these issues, we propose Hyper2Edge, an efficient and robust framework for trajectory similarity computation via representation learning. Our framework eliminates the reliance on multi-metric supervision by learning directly from Euclidean-based similarity labels; it also avoids repetitive encoding through a hierarchical trajectory representation learning framework and a novel

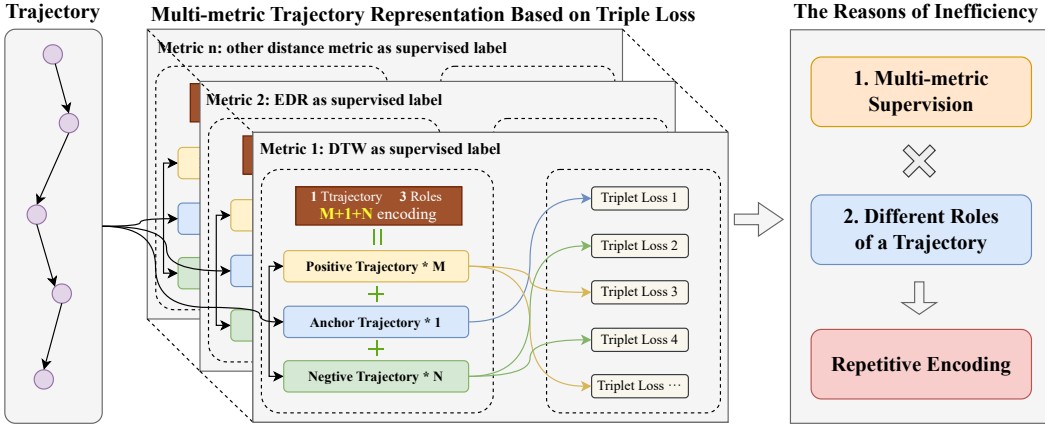

Figure 1: Two reasons lead to repetitive encoding: (i) Multi-distance metrics lead to fully-cycle training repetitively; (ii) The same trajectories as different roles need to be repetitive encoding due to the properties of triplet loss.

weighted top-$k$ InfoNCE loss. Specifically, Hyper2Edge first models trajectories as hyperedges within a hypergraph, offering distinct advantages over conventional graph-based methods. This approach provides two key benefits: (i) it preserves complete structural and sequential information without reducing trajectories to single nodes; (ii) it avoids the computationally expensive pairwise similarity calculations required for edge construction in traditional graphs. This is followed by a hierarchical trajectory representation learning architecture that captures both intra-trajectory patterns and inter-trajectory similarities. Finally, a weighted top-$k$ InfoNCE loss is introduced to overcome the limitations of triplet loss by emphasizing discrimination among the top-$k$ most similar trajectories. This design eliminates repetitive encoding of positive and negative samples, significantly improving training efficiency and representation robustness. The main contributions of this paper are summarized as follows:

- We propose a novel framework, Hyper2Edge, that learns trajectory representations for efficient and robust similarity computation by directly adopting Euclidean-based supervision and a non-repetitive encoding scheme.

- We devise a hierarchical trajectory representation learning architecture that models trajectories as hyperedges and employs node-hyperedge bidirectional message passing to jointly capture intra- and inter-trajectory patterns, enhanced by a weighted top-$k$ InfoNCE loss for local trajectory similarity learning and local structural consistency without repetitive encoding of samples.

- We conduct the experiments on two benchmark datasets. The results verify that the proposed method significantly outperforms the state-of-the-art baselines on the task of trajectory similarity computation.

## 2 PRELIMINARIES

**Definition 1** (*Trajectory*) Each trajectory $TR_i = \{p_1, p_2, \cdots, p_i, \cdots\}$ is a sequence of GPS points. A trajectory point $p_i = (lat_i, long_i)$ consists of latitude $lat_i$ and longitude $long_i$ of the vehicle's location.

**Definition 2** (*Spatial Token*) In ride-sharing scenario, a set of spatial tokens $STs = \{ST_1, ST_2, \cdots, ST_n\}$ is generated by applying the K-Means clustering algorithm to all trajectory points $\{p_1, p_2, \cdots, p_z\}$ within the region. The K-Means algorithm partitions the $z$ points into $n$ clusters by minimizing the within-cluster sum of squared distances, formalized as:

$$\underset{STs}{\arg\min} \sum_{j=1}^{n} \sum_{p_i \in \mathcal{C}_j} |p_i - ST_j|^2. \tag{1}$$

Each resulting spatial token $ST_j$ is represented by the centroid (mean geographic location) of its cluster $\mathcal{C}_j$, calculated as:

$$ST_j = \left( \frac{1}{|\mathcal{C}j|} \sum_{p_i \in \mathcal{C}_j} lat_i, \frac{1}{|\mathcal{C}j|} \sum_{p_i \in \mathcal{C}_j} long_i \right), \tag{2}$$

where $|\mathcal{C}_j|$ denotes the number of trajectory points in cluster $\mathcal{C}_j$.

**Definition 3** (*Tokenized Trajectory*) A trajectory $TR_i$ is transformed into a tokenized trajectory $TT_i$, defined as:

$$TT_i = \{ST_j \mid TR_i \text{ traverses } ST_j \text{ sequentially}\}, \tag{3}$$

where each $ST_j$ denotes a spatial token (with a unique global identifier $j$) that the trajectory $TR_i$ visits in chronological order.

**Problem Statement** (*Trajectory Representation Learning*) Given a collection of tokenized trajectory $TT$ and Euclidean distance $D_{\text{true}}(TT_i, TT_j)$, the trajectory representation problem is to learn a function $F$ which maps $TT$ into a $d$-dimensional vector. The goal of this problem is to reduce the discrepancy between the Euclidean distance provided by $D_{\text{true}}(TT_i, TT_j)$ and the similarity scores $||F(TT_i) - F(TT_j)||_2$, as quantified by the absolute difference $|||F(TT_i) - F(TT_j)||_2 - D_{\text{true}}(TT_i, TT_j)|$.

## 3 METHODOLOGY

This paper proposes Hyper2Edge, an efficient and robust trajectory representation learning framework for trajectory similarity computation, shown in Figure 2. The framework consists of three main components: (i) Hypergraph-Based Trajectory Modeling, which represents each tokenized trajectory as a hyperedge connecting multiple spatial tokens to avoid critical information loss; (ii) Hierarchical Trajectory Representation Learning, which comprises an initial embedding generation layer that capture sequential information of tokenized trajectories, a bidirectional trajectory encoding layer that employs multi-step node-hyperedge interaction to capture the inter-relationships between tokenized trajectories, and a trajectory representation decoding layer that produces final trajectory embeddings; (iii) Weighted Top-$k$ InfoNCE Loss, which optimizes the model without repetitively encoding positive and negative samples.

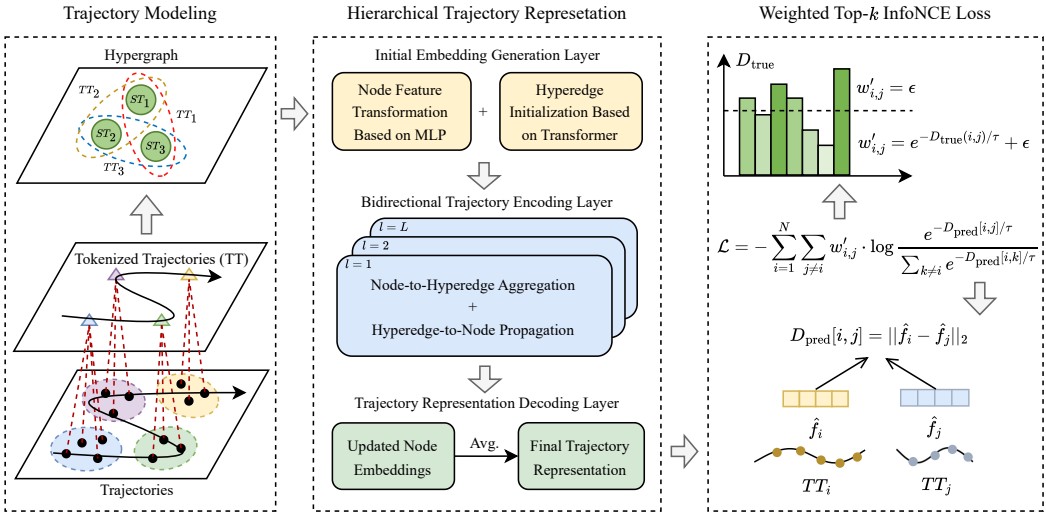

Figure 2: The framework of Hyper2Edge.

### 3.1 HYPERGRAPH-BASED TRAJECTORY MODELING

Unlike other methods Cheng et al. (2024) that construct traditional graphs, we construct hypergraphs. Formally, hypergraph Ding et al. (2020); Wang et al. (2024) can be defined as follows:

**Definition 4** (*Hypergraph*) A hypergraph is a generalized graph where each hyperedge can connect any number of nodes. Formally, it is defined as $\mathcal{G} = (\mathcal{V}, \mathcal{E})$, where $\mathcal{V}$ is the set of nodes and $\mathcal{E}$ is the set of hyperedges, with each $e \in \mathcal{E}$ satisfying $e \subseteq \mathcal{V}$.

**Definition 5** (*Incidence Matrix*) The incidence matrix of a hypergraph $\mathcal{G} = (\mathcal{V}, \mathcal{E})$ is a binary matrix $H \in \{0,1\}^{|\mathcal{V}| \times |\mathcal{E}|}$, where each entry $H(v, e) = 1$ if node $v \in \mathcal{V}$ belongs to hyperedge $e \in \mathcal{E}$, and 0 otherwise.

In this scenario, the vertex set $\mathcal{V}$ is defined as all spatial tokens, while the hyperedge set $\mathcal{E}$ comprises all tokenized trajectories. The features of vertex $X = [\mathrm{x}_1, \mathrm{x}_2, \cdots, \mathrm{x}_n]^{\mathrm{T}} \in \mathbb{R}^{n \times 2}$, where $n$ is the number of spatial tokens, are derived from Equation 2. Each hyperedge symbolizes a trajectory, linking the multiple spatial tokens visited by that trajectory in chronological sequence. The incidence relationship between vertices and hyperedges can be succinctly captured by an incidence matrix $H \in \{0,1\}^{n \times t}$, where each entry $H_{ij}$ indicates whether vertex $v_i$ is part of hyperedge $e_j$, and $t$ is the number of tokenized trajectories. As a result, a hypergraph $\mathcal{G} = (H, X)$ is established. Therefore, the problem of tokenized trajectory representation becomes equivalent to the problem of hyperedge representation.

## 3.2 Hierarchical Trajectory Representation Learning

After converting tokenized trajectories into hyperedges, it is necessary to encode the resulting hypergraph to obtain the final hyperedge representations, which correspond to the trajectory embeddings. To this end, we design a hierarchical trajectory representation learning architecture.

### 3.2.1 Initial Embedding Generation Layer

To adaptively learn task-optimized representations of tokenized trajectories, we initialize the nodes and hyperedges, enabling the model to identify and enhance the most critical features of both.

**Node Feature Transformation.** The initial representation $u_i$ for the $i$-th node is obtained by projecting its raw feature vector $\mathbf{x}_i$ into a $d$-dimensional latent space using a linear layer followed by ReLU activation:

$$u_i = \mathrm{ReLU}\left(\mathbf{x}_i W_1 + b\right) \in \mathbb{R}^d, \tag{4}$$

where $W_1 \in \mathbb{R}^{2 \times d}$ and $b \in \mathbb{R}^d$ are learnable parameters.

**Order-aware Hyperedge Initialization.** A simple way to represent hyperedges is through the averaging of their node features. However, this approach ignores essential sequence information. To address this, we employ a Transformer encoder that directly operates on the original node features while explicitly incorporating chronological order.

For a hyperedge $e_i$ containing a set of nodes, each node's representation is formed by concatenating its original feature vector $\mathrm{x}_k$ and learnable positional encoding $p_k$. These enhanced node representations are then processed by a Transformer encoder to capture both local and global contextual relationships:

$$Z_i = \mathrm{TransformerEncoder}\left(\left[(\mathrm{x}_1 \parallel p_1), (\mathrm{x}_2 \parallel p_2), \ldots, (\mathrm{x}_m \parallel p_m)\right]\right) \in \mathbb{R}^{m \times d}, \tag{5}$$

where $m$ denotes the number of nodes in the hyperedge, and $\parallel$ represents concatenation. We use the output representation at the final sequence position as the initial representation of hyperedge $e_i$:

$$f_i = Z_i[-1, :] \in \mathbb{R}^d. \tag{6}$$

This representation aggregates information across the entire sequence through the Transformer's self-attention mechanism, effectively capturing trajectory sequential information.

### 3.2.2 Bidirectional Trajectory Encoding Layer

Tokenized trajectories exhibit complex structural properties: each connects multiple spatial tokens, while a single spatial token may belong to multiple tokenized trajectories. To effectively capture structural information, we propose a Bidirectional Trajectory Encoding Layer (BTELayer). This layer enables each tokenized trajectory to attend to both its own spatial tokens and other trajectories that share its tokens.

**Bottom-Up: Node-to-Hyperedge Aggregation.** In the bottom-up phase, each hyperedge aggregates information from its constituent nodes. Formally, for hyperedge $e_i$ containing a set of nodes $\mathcal{V}_i$, we compute its updated representation as:

$$f_i^{\text{agg}} = \frac{1}{|\mathcal{V}_i|} \sum_{v_k \in \mathcal{V}_i} u_i W_2 \in \mathbb{R}^d, \tag{7}$$

where $W_2 \in \mathbb{R}^{d \times d}$ is a learnable projection matrix that aligns node features with the hyperedge representation space.

We employ a residual gating mechanism to avoid losing the original features of tokenized trajectories and to allow the network to adaptively control information flow:

$$g_i^{\text{edge}} = \sigma \left( W_{\text{gate}}^{\text{edge}}[f_i \parallel f_i^{\text{agg}}] + b_{\text{gate}}^{\text{edge}} \right) \in \mathbb{R}^d, f_i' = \text{LayerNorm} \left( f_i + g_i^{\text{edge}} \odot f_i^{\text{agg}} \right) \in \mathbb{R}^d, \tag{8}$$

where $W_{\text{gate}}^{\text{edge}} \in \mathbb{R}^{d \times 2d}$ and $b_{\text{gate}}^{\text{edge}} \in \mathbb{R}^d$ are learnable parameters, $\sigma$ denotes the sigmoid activation function, $\parallel$ represents concatenation, and $\odot$ denotes element-wise multiplication.

**Top-Down: Hyperedge-to-Node Propagation.** Different tokenized trajectories may pass through the same spatial tokens, suggesting that these trajectories can share latent spatial or behavioral similarities. To effectively capture such inter-trajectory relationships, we introduce Hyperedge-to-Node propagation that propagates representations of hyperedge back to their corresponding nodes.

In this phase, each node aggregates information from hyperedges it participates in. Let $v_i$ denote the $i$-th node, and $\mathcal{E}_i$ be the set of hyperedges it belongs to. Each hyperedge $e_k \in \mathcal{E}_i$ is initially represented as $f_k'$, and is transformed via a learnable linear projection:

$$u_i^{\text{agg}} = \frac{1}{|\mathcal{E}_i|} \sum_{e_k \in \mathcal{E}_i} f_k' W_3 \in \mathbb{R}^d, \tag{9}$$

where $W_3 \in \mathbb{R}^{d \times d}$ is a learnable projection matrix.

Similar to the bottom-up phase, the node representation is then updated via a residual gating mechanism:

$$g_i^{\text{node}} = \sigma \left( W_{\text{gate}}^{\text{node}}[u_i \parallel u_i^{\text{agg}}] + b_{\text{gate}}^{\text{node}} \right) \in \mathbb{R}^d, u_i' = \text{LayerNorm} \left( u_i + g_i^{\text{node}} \odot u_i^{\text{agg}} \right) \in \mathbb{R}^d, \tag{10}$$

where $W_{\text{gate}}^{\text{node}} \in \mathbb{R}^{d \times 2d}$ and $b_{\text{gate}}^{\text{node}} \in \mathbb{R}^d$ are learnable parameters, $\sigma$ denotes the sigmoid activation function, $\parallel$ represents concatenation, and $\odot$ denotes element-wise multiplication.

This top-down propagation enables each node to integrate structural and sequential patterns from the hyperedges, effectively enriching its representation to capture latent inter-trajectory similarities.

**Iterative Multi-Scale Representation Refinement.** The above bidirectional process is repeated for $L$ layers, allowing information to propagate through multiple hops in the hypergraph:

$$U^{(l)}, F^{(l)} = \text{BTELayer} \left( U^{(l-1)}, F^{(l-1)}, H \right), \tag{11}$$

where $U^{(l)} \in \mathbb{R}^{|\mathcal{V}| \times d}$ and $F^{(l)} \in \mathbb{R}^{|\mathcal{E}| \times d}$ denote the node and hyperedge representations at layer $l$. This iterative refinement captures multi-scale structural patterns and enhances the expressiveness of both node and hyperedge representations.

### 3.2.3 TRAJECTORY REPRESENTATION DECODING LAYER

In the final stage, we generate the semantically enriched representation for each hyperedge by aggregating the updated node embeddings. Specifically, for a hyperedge $e_i$ containing a set of nodes $\mathcal{V}_i$, its final representation $\hat{f}_i$ is computed as follow:

$$\hat{f}_i = \frac{1}{|\mathcal{V}i|} \sum_{v_k \in \mathcal{V}_i} u_k' \in \mathbb{R}^d. \tag{12}$$

By aggregating the refined node embeddings, this decoding layer generates the final trajectory (hyperedge) representations, which retain original features while integrating complex patterns learned by the network. This process yields highly expressive embeddings that are optimized for trajectory similarity computation.

### 3.3 WEIGHTED TOP-$k$ INFONCE LOSS

Based on insights from prior work Yao et al. (2019); Yang et al. (2021; 2022), we identify a key limitation in prevailing triplet loss-based methods: their myopic focus on a handful of positive and negative samples restricts each trajectory from perceiving its position within the global similarity structure. To overcome this, we design a novel loss that enables each tokenized trajectory to strategically enhance the discrimination among its top-$k$ most similar neighbors for robust local structure preservation. We first construct an initial similarity weighted matrix derived from the ground-truth Euclidean distance matrix $D_{\text{true}}$:

$$w_{i,j} = \frac{e^{-D_{\text{true}}[i,j]/\tau}}{\sum_{k \neq i} e^{-D_{\text{true}}[i,k]/\tau}}, \tag{13}$$

where $\tau$ is a temperature coefficient that controls the smoothness of the similarity distribution.

Although this design captures the overall similarity structure among tokenized trajectories, it may not sufficiently emphasize strong local correlations that are critical for downstream tasks such as trajectory similarity computation. To address this, we introduce a weighted top-$k$ enhancement mechanism that explicitly reinforces the influence of local neighbors. Specifically, for each tokenized trajectory $TT_i$, we select its top-$k$ nearest neighbor set $\mathcal{N}_i$ based on $D_{\text{true}}$ and redefine the weighted matrix:

$$w'_{i,j} = \begin{cases} e^{-D_{\text{true}}(i,j)/\tau} + \epsilon, & j \in \mathcal{N}_i \\ \epsilon, & j \notin \mathcal{N}_i \end{cases}, \tag{14}$$

where $\epsilon$ is a very small positive number to ensure numerical stability.

Subsequently, each row is normalized to ensure that the weighted distribution of each sample satisfies the probability constraints:

$$w'_{i,j} \leftarrow \frac{w'_{i,j}}{\sum_{n=1}^{N} w'_{i,n}}. \tag{15}$$

Building upon this, and inspired by InfoNCE Oord et al. (2018), we propose a weighted top-$k$ InfoNCE loss, defined as follows:

$$\mathcal{L} = -\sum_{i=1}^{N} \sum_{j \neq i} w'_{i,j} \cdot \log \frac{e^{-D_{\text{pred}}[i,j]/\tau}}{\sum_{k \neq i} e^{-D_{\text{pred}}[i,k]/\tau}}, \tag{16}$$

where $D_{\text{pred}}[i,j]$ denotes the predicted distance between tokenized trajectories $TT_i$ and $TT_j$, computed as the Euclidean distance between their corresponding representation vectors $\hat{f}_i$ and $\hat{f}_j$, i.e:

$$D_{\text{pred}}[i,j] = ||\hat{f}_i - \hat{f}_j||_2. \tag{17}$$

By minimizing this loss, the model is able to fit the true similarity distribution between tokenized trajectories in the feature space: making similar tokenized trajectories close together and dissimilar tokenized trajectories far away in the representation space, thus providing a more discriminative feature representation for trajectory similarity computation. By avoiding repetitive encoding of tokenized trajectories for triplet loss computation, this method significantly improves efficiency.

### 3.4 COMPLEXITY ANALYSIS

Based on the proposed framework, we analyze the computational complexity of Hyper2Edge. The overall time complexity is dominated by the Transformer-based hyperedge initialization and the iterative bidirectional encoding process. For a hypergraph with $|\mathcal{V}|$ nodes (spatial tokens) and $|\mathcal{E}|$ hyperedges (tokenized trajectories), the Transformer encoder processes each hyperedge in $O(m^2 \cdot d)$ time where $m$ is the number of nodes in a hyperedge after spatial clustering. The bidirectional trajectory encoding layer performs message passing in $O(L \cdot (|\mathcal{V}| + |\mathcal{E}|) \cdot d^2)$ time over $L$ layers. Thus, the overall time complexity of the proposed Hyper2Edge framework is $O(|\mathcal{E}| \cdot m^2 \cdot d + L \cdot (|\mathcal{V}| + |\mathcal{E}|) \cdot d^2)$, which scales linearly with the number of spatial tokens and tokenized trajectories. A detailed comparative analysis of complexity is provided in the appendix C.

# 4 EXPERIMENT

We systematically evaluate Hyper2Edge through five experiments on two real-world public datasets, aiming to address the following research questions: (1) Does Hyper2Edge correctly represent both the tokenized trajectories themselves and the relationships between them? (2) How robust is Hyper2Edge across different distance metrics? (3) Does Hyper2Edge truly enhance computational efficiency? (4) How does each component that we design contribute to the model performance? (5) How sensitive is Hyper2Edge to its parameters? (6) Does Hyper2Edge produce human-interpretable results? (7) Can Hyper2Edge generalize to spatio-temporal contexts? (8) How does Hyper2Edge perform on large datasets?

## 4.1 EXPERIMENTAL SETTINGS

We briefly introduce the experimental settings below. The detailed experimental settings can be found in the Appendix D.1. **Datasets.** We experiment on two real-world trajectory datasets: GeoLife Zheng et al. (2010) and Porto O'Connell et al. (2015). **Experimental Baselines.** We evaluate the proposed Hyper2Edge against several prominent TRL methods from recent years: (i) Unsupervised methods: t2vec Li et al. (2018), CL-Tsim Deng et al. (2022) and HHL-Traj Cao et al. (2024); (ii) Supervised methods: NeuTraj Yao et al. (2019), TrajGAT Yao et al. (2022), TrajCL Chang et al. (2023), and SIMformer Chuang et al. (2024). **Evaluation Metrics.** In the top-$k$ trajectory similarity search task, we evaluate performance using Hit Rate (HR) and Recall (R), reported as HR@1, HR@5, HR@10, R1@5, R5@10. Higher values for these metrics indicate greater accuracy.

## 4.2 OVERALL PERFORMANCE (RQ1)

We compare performance of Hyper2Edge against baseline methods on the Geolife and Porto datasets. The results in Table 1 clearly show that Hyper2Edge achieves state-of-the-art performance across both datasets and all evaluation metrics on Euclidean distance. Similar trends are observed on the DTW and ERP distance, detailed in Appendix D.2. On GeoLife, Hyper2Edge outperforms the strongest baseline SIMformer by margins of +2.5% in HR@1 and +4.0% in HR@10, while on Porto it surpasses CL-Tsim by +5.0% in HR@1 and +8.4% in HR@10. It is worth noting that although HHL-Traj also employs hypergraph encoding, its task involves finding the second half of its own trajectory based on the first half rather than learning the relationship between trajectories. Consequently, its metrics for this task are nearly zero. These consistent improvements validate that Hyper2Edge not only learns expressive trajectory representations but also captures the relational order and structure among trajectories through its hypergraph-based modeling and hierarchical representation learning. Furthermore, the strong gains in recall metrics (R1@5 and R5@10) highlight the effectiveness of the proposed weighted top-$k$ InfoNCE loss in aligning the learned embedding space with Euclidean-based similarity, enabling more accurate neighborhood retrieval. Together, these results confirm that Hyper2Edge simultaneously preserves both intra-trajectory patterns and inter-trajectory relationships.

Table 1: Performance of top-$k$ trajectory similarity search on Euclidean distance.

| Dataset | Method | Ref. | HR@1 | HR@5 | HR@10 | R1@5 | R5@10 |
|---------|--------|------|------|------|-------|------|-------|
| GeoLife | t2vec | ICDE'18 | 11.77% | 15.40% | 18.98% | 23.96% | 23.25% |
| | CL-Tsim | CIKM'22 | 12.08% | 18.44% | 23.41% | 27.71% | 28.29% |
| | HHL-Traj | CIKM'24 | 0.00% | 0.06% | 0.18% | 0.00% | 0.13% |
| | NeuTraj | ICDE'19 | 6.15% | 12.21% | 15.92% | 16.77% | 19.13% |
| | TrajGAT | KDD'22 | 15.21% | 26.04% | 31.82% | 37.50% | 39.42% |
| | TrajCL | ICDE'23 | 4.38% | 12.08% | 19.73% | 24.45% | 27.71% |
| | SIMformer | VLDB'24 | 21.04% | 26.56% | 30.19% | 42.08% | 38.54% |
| | Hyper2Edge | Ours | **23.54%** | **29.67%** | **35.53%** | **42.92%** | **44.25%** |
| Porto | t2vec | ICDE'18 | 4.52% | 5.01% | 5.28% | 8.94% | 6.91% |
| | CL-Tsim | CIKM'22 | 15.10% | 19.71% | 21.40% | 31.05% | 27.94% |
| | HHL-Traj | CIKM'24 | 0.00% | 0.05% | 0.11% | 0.00% | 0.12% |
| | NeuTraj | ICDE'19 | 2.55% | 5.00% | 6.26% | 7.52% | 7.40% |
| | TrajGAT | KDD'22 | 11.14% | 17.92% | 21.75% | 26.30% | 27.10% |
| | TrajCL | ICDE'23 | 0.20% | 17.42% | 22.95% | 24.63% | 37.22% |
| | SIMformer | VLDB'24 | 9.73% | 14.47% | 15.74% | 21.95% | 20.48% |
| | Hyper2Edge | Ours | **20.14%** | **27.18%** | **29.78%** | **41.46%** | **37.45%** |

## 4.3 IN-DEPTH ANALYSIS

**Cross-Distance Metric Robustness Study (RQ2).** In this analysis, we define two training strategies: (i) Self-distance, where the model is trained and evaluated on the same benchmark metric (e.g., DTW), and (ii) Cross-distance, where the model is trained using Euclidean distance supervision and evaluated on various benchmark metrics. It is worth noting that Discrete Fréchet, DTW and ERP are classic trajectory similarity measures specifically designed to handle variations in sampling rates, temporal scaling, and noise. Figure 3 evaluates the robustness of Hyper2Edge under different distance metrics from the GeoLife dataset; similar trends were observed on the Porto dataset and are detailed in Appendix D.3. Our results indicate that the cross-distance strategy performs comparably to or even better than the self-distance strategy when assessed on the same metric. Therefore, the robustness of Hyper2Edge is evidenced by its consistent performance across various metrics (Discrete Fréchet, DTW, ERP), showing that its effectiveness is not tightly bound to a particular metric supervision. Notably, training only once with Euclidean distance is sufficient to capture essential trajectory similarity patterns across different metrics, thereby eliminating the need for redundant training with multiple metric supervision and achieving the "train once, use everywhere" paradigm.

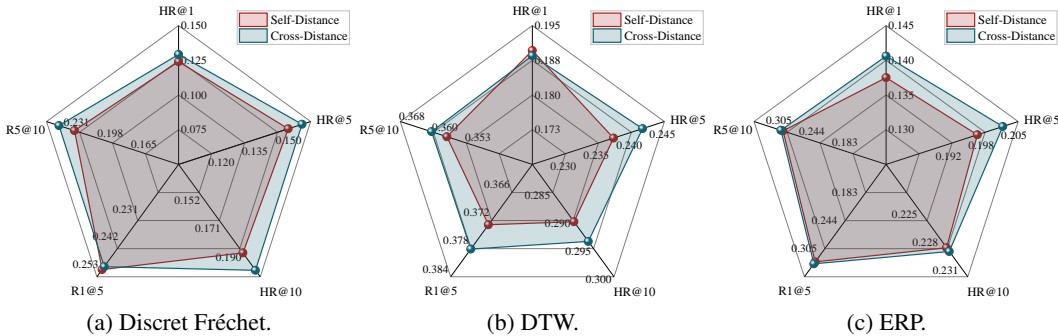

(a) Discret Fréchet.  (b) DTW.  (c) ERP.

Figure 3: Cross-distance metic robustness on GeoLife dataset.

**Efficiency Evaluation (RQ3).** We design three experiments to analyze efficiency of Hyper2Edge. First, we compare each epoch time of Hyper2Edge with all baseline methods on GeoLife and Porto dataset, as shown in Table 2. Although SIMformer shows the fastest per-epoch time, Hyper2Edge maintains a compelling balance of competitive efficiency and superior performance, as established in our overall performance analysis (RQ1, Section 4.2). Second, we conduct a fine-grained efficiency analysis, and the detailed breakdown per training epoch is provided in Table 3. Third, we analyze loss efficiency that replaces the weighted top-$k$ InfoNCE loss with a standard triplet loss, as shown in Table 4. The results show that the training-time reduction is a direct result of our methodological contributions: (i) Zero Sampling Overhead: The design of our framework inherently avoids the explicit sampling of positive/negative pairs; (ii) Single Encoding Pass: The key innovation is that each trajectory requires only a single encoding pass, eliminating repetitive and costly graph operations for the same trajectory; (iii) Efficient Loss Design: Our proposed loss function strengthens trajectory proximity relations while eliminating the need for explicit positive/negative pair sampling and encoding, significantly reducing overall runtime; (iv) Fair Comparison: The gains are thus derived from a fundamental architectural efficiency, not from an unbalanced experimental setup.

Table 2: Each epoch time (s) comparison on GeoLife and Porto datasets.

| Method | GeoLife | Porto |
|---|---|---|
| t2vec | 35 | 36 |
| NeuTraj | 38 | 123 |
| TrajGAT | 283 | 515 |
| CL-Tsim | 4 | 17.6 |
| TrajCL | 120 | 524 |
| HHL-Traj | 5 | 11 |
| SIMformer | **3** | **5** |
| Hyper2Edge | 4 | 9 |

Table 3: Fine-grained efficiency study. **Note**, Encoding Passes = (Sampling Overhead + Graph Ops + Weighted Top-$k$ InfoNCE Loss + Gradient Update).

| Each Epoch Time (s) | GeoLife | Porto |
|---|---|---|
| Sampling Overhead | 0 | 0 |
| Graph Ops | 0.63 | 1.11 |
| Weighted Top-$k$ InfoNCE Loss | 0.02 | 0.05 |
| Gradient update | 0.98 | 1.7 |
| Encoding Passes | 1.65 | 2.86 |
| Evaluation | 1.93 | 3.8 |
| Validation | 1.05 | 2.35 |
| Total Time | 4 | 9 |

Table 4: Efficiency study with triplet loss replacing weighted top-$k$ InfoNCE loss.

| Each Epoch Time (s) | GeoLife | Porto |
|---|---|---|
| Weighted Top-k InfoNCE Loss | **4** | **9** |
| Triplet Loss | 6 | 12 |

**Ablation Study (RQ4).** To further verify the effectiveness of each component in Hyper2Edge, we compare Hyper2Edge with the following variants: (i) *w/o* Order: the initial hyperedge representation is directly obtained by MLP. (ii) *w/o* Node-to-Edge: the hyperedge representation is derived only from the raw features of spatial tokens, without leveraging the initial node features. (iii) *w/o* Edge-to-Node: the hyperedge-to-node propagation module is removed, which prevents the model from capturing similarity features among hyperedges that share common nodes. (iv) *w/o* Weighted Top-$k$ InfoNCE Loss: this loss is replaced by Mean Square Error (MSE) loss.

We report results on the GeoLife and Porto datasets in Figure 4, showing HR@1 and HR@5 for brevity, as the remaining three metrics exhibit similar trends and are provided in the Appendix D.4. We can observe the following: (i) by comparing *w/o* order with Hyper2Edge, we find that the order-aware hyperedge inititalization can capture the trajectory's own motion patterns and sequential properties; (ii) by comparing *w/o* Node-to-Edge with Hyper2Edge, we find that incorporating initial node features is essential for learning more informative tokenized trajectory representation; (iii) by comparing *w/o* Edge-to-Node with Hyper2Edge, we find that Hyperedge-to-Node prpogation can capture the similarity characteristics among tokenized trajectories; (iv) by comparing *w/o* Weighted Top-$k$ InfoNCE Loss with Hyper2Edge, we find that weighted Top-$k$ InfoNCE loss can bring the trajectory distribution closer to the true distribution; (v) Hyper2Edge outperforms all variants with ablation, which proves the effectiveness of the proposed method.

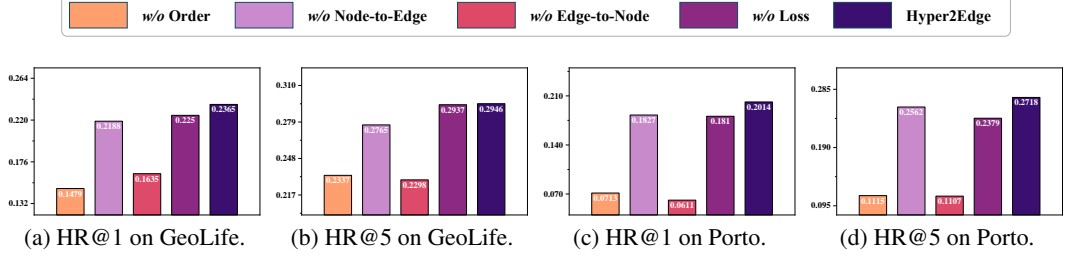

(a) HR@1 on GeoLife.    (b) HR@5 on GeoLife.    (c) HR@1 on Porto.    (d) HR@5 on Porto.

Figure 4: Ablation results by removing components on HR@1 and HR@5.

Additionally, we perform ablation study by replacing the weighted top-$k$ InfoNCE loss with a standard triplet loss. The results in Table 5 consistently show that weighted top-$k$ InfoNCE loss achieves superior performance across all evaluation metrics on both GeoLife and Porto datasets. This confirms the effectiveness of our proposed loss design over a standard triplet loss baseline.

**Parameter Sensitivity Study (RQ5).** To assess the sensitivity of Hyper2Edge, we analyze four key hyperparameters: the number of spatial tokens $n$, the hidden dimension $d$, the temperature coefficient $\tau$, and the top-$k$ quantity. The spatial tokens are formed using K-Means, and $n$ denotes their number. The hypergraph sparsity and tokenization choices (e.g., clustering granularity) are both controlled by the number of cluster $n$. The hidden layer dimension $d$ represents the output

Table 5: Ablation results with triplet loss replacing weighted top-$k$ InfoNCE loss.

| Dataset | Ablation Loss | HR@1 | HR@5 | HR@10 | R1@5 | R5@10 |
|---------|---------------|------|------|-------|------|-------|
| GeoLife | Weighted Top-$k$ InfoNCE Loss | **23.54%** | **29.67%** | **35.53%** | **42.92%** | **44.25%** |
|         | Triplet Loss | 17.92% | 29.13% | 35.25% | 40.63% | 43.87% |
| Porto   | Weighted Top-$k$ InfoNCE Loss | **20.14%** | **27.18%** | **29.78%** | **41.46%** | **37.45%** |
|         | Triplet Loss | 15.55% | 20.85% | 23.07% | 32.07% | 29.02% |

dimension of the representation vector. The temperature coefficient $\tau$ represents the softness of the model's output probability distribution. The number of top-$k$ represents the number of nearest neighbors that are mainly fitted when the loss is fitted to a distribution. Figure 5 presents the analysis results on the GeoLife dataset, while Appendix D.5 reports similar trends on the Porto dataset. As shown, HR@1 and HR@5 remain stable across a wide range of parameters, indicating robustness of Hyper2Edge to hyperparameter selection.

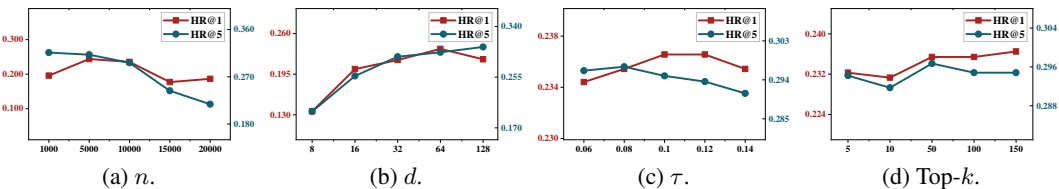

(a) $n$.    (b) $d$.    (c) $\tau$.    (d) Top-$k$.

Figure 5: Effect of different $n$, $d$, $\tau$ and top-$k$ on GeoLife. The $y$-axis represents hit rate and the $x$-axis is the different hyper-parameter values.

**Case Study (RQ6).** To analyze the interpretability of Hyper2Edge, we compare the top-2 nearest neighbors of a query trajectory from the ground-truth Euclidean distance against those from Hyper2Edge. The results show that the matching outcomes generated by Hyper2Edge align almost perfectly with the ground truth in terms of start points, end points, and paths. This validates that Hyper2Edge learns human-perceivable semantic patterns.

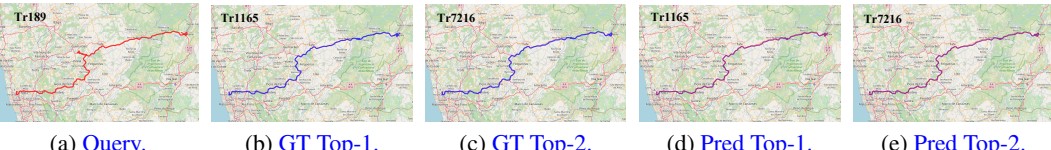

(a) Query.    (b) GT Top-1.    (c) GT Top-2.    (d) Pred Top-1.    (e) Pred Top-2.

Figure 6: Spatial comparison with query of top-1 and top-2 similar trajectories between the ground truth (GT) and Hyper2Edge (Pred).

**Generalization Study in Spatio-temporal Contexts (RQ7).** We evaluate Hyper2Edge learned spatio-temporal representations against a spatio-temporal ground truth distance. As shown in Appendix D.6, Hyper2Edge performs nearly identically under both spatial and spatio-temporal evaluation, proving its inherent capability to learn effective representations even when temporal factors are considered. This suggests that Hyper2Edge enable to learn robust patterns that generalize to spatio-temporal contexts.

**Scalability Study (RQ8).** We perform top-$k$ trajectory similarity search task on the larger-scale Porto dataset. The results (shown in Appendix D.7) show that performance of Hyper2Edge remains stable and consistent with our initial findings in our overall performance analysis (RQ1, Section 4.2).

## 5 CONCLUSION

We propose Hyper2Edge, a novel representation learning framework that improves efficiency and robustness of trajectory similarity computation. Hyper2Edge employs hypergraph-based trajectory modeling to represent trajectories as hyperedges to preserve essential information. Its hierarchical trajectory encoding architecture includes order-aware embedding initialization, bidirectional node-hyperedge message passing to capture inter-trajectory relationships, and a decoding layer that outputs discriminative embeddings. A weighted top-$k$ InfoNCE loss further enhances local similarity learning without sample re-encoding. Evaluations on two public benchmarks show that Hyper2Edge outperforms state-of-the-art methods. A promising future direction is developing learnable metric functions that can autonomously adapt to different trajectory patterns. Such an approach could potentially achieve more accurate similarity computation especially under challenging conditions.

ETHICS STATEMENT

Our research aims to improve the efficiency and robustness of trajectory similarity computation via representation learning. This work is technical and practical in nature, with potential applications in intelligent transportation and location-based services. We have carefully considered possible societal implications and identify no immediate ethical risks or harmful consequences resulting from our methodology. We support the ethical use of our research and advocate for its responsible implementation in real-world systems.

REPRODUCIBILITY STATEMENT

All the results in this work are reproducible. We provide all the necessary code to replicate our results in an anonymous GitHub repository (https://anonymous.4open.science/r/Hyper2Edge-3D2B). The repository includes code of data preprocessing and model, environment configurations, run scripts, and other relevant materials. We discuss the detailed experimental settings in Section D.1.

LLM USAGE

In this study, large language models (LLMs) are employed to enhance the linguistic quality and stylistic refinement of the text. Their application is strictly limited to polishing language expression and does not involve content generation or substantive analysis.

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

## A  RELATED WORK

**Non-learning-based Methods.** Non-learning-based approaches, such as Discrete Fréchet Alt & Godau (1995), DTW Rakthanmanon et al. (2012), and ERP Chen & Ng (2004), typically rely on pairwise trajectory comparisons using Euclidean distance and predefined rules to identify optimal matches. These methods often struggle to generalize across diverse scenarios due to their inflexible design, which is tightly coupled with specific distance metrics. Additionally, they suffer from a time complexity of $O(z^2)$, where $z$ is the number of trajectory points, making them inefficient.

**Learning-based Methods with Road Network.** To overcome these limitations, learning-based methods incorporate road network data to enhance trajectory similarity measurement. One line of work, including ST2Vec Fang et al. (2022), PT2Vec Li et al. (2023), and START Jiang et al. (2023), aligns trajectories to road segments but may lose original spatial information when matches are incomplete. Another category combines trajectory and road network features: GTS Han et al. (2021) and GRLSTM Zhou et al. (2023) model their interactions via graph structures, FEST Zhou et al. (2025a) uses road networks as auxiliary features, and JCLRNT Mao et al. (2022) employs contrastive learning to maximize mutual information between the two. JGRM Ma et al. (2024) processes trajectories and road networks separately before fusion. Recently, Luo et al. (2024) also uses causal intervention to reduce environmental bias in trajectory representations.

**Learning-based Methods without Road Network.** Alternatively, some methods avoid road networks to better capture intrinsic motion patterns. Methods such as t2vec Li et al. (2018), CL-Tsim Deng et al. (2022), and E2DTC Fang et al. (2021) use sequence-to-sequence models with spatial-aware loss functions. KGTS Chen et al. (2024) incorporates knowledge graphs and prompt learning. Other approaches integrate inter-trajectory relations using distance metrics like DTW Rakthanmanon et al. (2012) or Hausdorff Alt (2009) as supervision—e.g., NeuTraj Yao et al. (2019) and Aries Feng et al. (2022) learn spatial embeddings via attention mechanisms, while Traj2SimVec Zhang et al. (2020) and TMN Yang et al. (2022) improve matching efficiency and robustness. TrajCL Chang et al. (2023) incorporates structural features, and TrajGAT Yao et al. (2022) and STTraj2Vec Zhu et al. (2024) use hierarchical or graph-based representations, though often at increased computational cost. Efflex Cheng et al. (2024) views trajectories as nodes, inter-trajectory similarities as connecting edges of nodes, and uses a graph model for node representation, which discards internal structure and creates a circular dependency on precomputed similarities.

## B  DIFFERENCES FROM KGTS

There are superficial similarities between Hyper2Edge and KGTS Chen et al. (2024) in 'using contrastive learning', but Hyper2Edge proposes an entirely novel solution driven by fundamentally different motivations: (i) Paradigm: KGTS employs a two-stage pipeline of 'grid → trajectory', whereas Hyper2Edge performs end-to-end hypergraph learning where 'trajectories serve as hyperedges'—a foundational architectural innovation; (ii) Core problem focus: We concentrate on resolving the 'repetitive encoding' efficiency bottleneck unaddressed by KGTS, rather than its emphasis on 'unsupervised label generation'; (iii) Technical approach: We introduce a bidirectional node-hyperedge message passing mechanism absent in KGTS to explicitly model inter-trajectory relationships, and design a weighted top-k mechanism to optimize supervised learning based on Euclidean distance. Therefore, Hyper2Edge delivers substantive innovations distinct from KGTS in problem definition, core architecture, and technical details.

## C  COMPLEXITY ANALYSIS

We provide a direct comparison with baseline methods in Table 6. Although most baselines exhibit complexities that are linear or quadratic in trajectory length $l$ (e.g., $O(|\mathcal{E}| \cdot l^2)$ for TrajCL and SIMformer), the complexity of Hyper2Edge is independent of the raw trajectory length. This is because our method operates on the tokenized trajectory representation, where the key parameter $m$ (the maximum number of tokens per trajectory) is typically much smaller and more stable than $l$ (the original number of points per trajectory), especially after spatial clustering. This analysis theoretically confirms that Hyper2Edge avoids the computational bottlenecks associated with long, raw trajectories and achieves superior scalability. Meanwhile, although the theoretical time com-

plexity of baseline methods is the time required to encode $|\mathcal{E}|$ trajectories, they actually encode the same trajectory multiple times as positive and negative samples for other trajectories to utilize triplet loss. Consequently, the encoding time of these methods exceeds the theoretical estimate, whereas Hyper2Edge strictly adheres to the theoretically analyzed time of encoding.

Table 6: Complexity Analysis. **Note**, $|\mathcal{E}|$ is the number of trajectories, $l$ is the original number of points per trajectory, $m$ is the number of spatial tokens per trajectory after spatial clustering and $|\mathcal{V}|$ is the total number of spatial tokens.

| Method | Complexity |
|--------|-----------|
| t2vec | $O(|\mathcal{E}| \cdot l)$ |
| CL-Tsim | $O(|\mathcal{E}| \cdot l)$ |
| HHL-Traj | $O(|\mathcal{E}| \cdot l)$ |
| NeuTraj | $O(|\mathcal{E}| \cdot l)$ |
| TrajGAT | $O(|\mathcal{E}| \cdot l \cdot log(l))$ |
| TrajCL | $O(|\mathcal{E}| \cdot l^2)$ |
| SIMformer | $O(|\mathcal{E}| \cdot l^2)$ |
| Hyper2Edge | $O(|\mathcal{E}| \cdot m^2 + |\mathcal{E}| + |\mathcal{V}|)$ |

## D    EXPERIMENT

### D.1    EXPERIMENTAL SETTINGS

**Datasets.** We experiment on two real-world trajectory datasets, i.e., GeoLife[1] Zheng et al. (2010) and Porto[2] O'Connell et al. (2015), which are widely used by TRL studies Li et al. (2018); Yao et al. (2019); Zhang et al. (2020); Yang et al. (2021; 2022); Yao et al. (2022); Jiang et al. (2023); Chang et al. (2023); Chen et al. (2024); Zhou et al. (2025b). The proportions of training and testing data are set to [0.8, 0.2] for both datasets. Following prior studies, we preprocess the GeoLife and Porto datasets by retaining trajectories with 50 to 200 points for GeoLife (representing short trajectories) and 200 to 300 points for Porto (representing long trajectories). The details of the two datasets are shown in Table 7.

Table 7: Statistical information of the two datasets.

|         | Trajs Number | Points Number | Min. Length | Max. Length | Avg. Length |
|---------|--------------|---------------|-------------|-------------|-------------|
| GeoLife | 4,769        | 526,640       | 50          | 200         | 109.81      |
| Porto   | 8,839        | 2,114,447     | 200         | 300         | 239.22      |

**Experimental Baselines.** We evaluate the proposed Hyper2Edge against several prominent TRL methods from recent years. For all baseline methods, we use the officially released code and default parameters.

(i) Unsupervised methods: This classification does not employ distance metrics as supervised labels for trajectory representation.

- t2vec Li et al. (2018): This method is based on the Seq2Seq model for trajectory similarity, which optimizes the error between the representation vector of a trajectory and the representation vector obtained when noise perturbation is added to the trajectory.
- CL-Tsim Deng et al. (2022): It leverages contrastive learning with point down-sampling and distorting augmentations to learn consistent trajectory representations for efficient and robust trajectory similarity computation.

---

[1]https://www.microsoft.com/en-us/download/details.aspx?id=52367

[2]https://www.kaggle.com/competitions/pkdd-15-predict-taxi-service-trajectory-i

- HHL-Traj Cao et al. (2024): It is a hypergraph hashing learning framework for encoding trajectories, with the learning objective being that the first half of a trajectory can locate its second half.

(ii) Supervised methods: This classification employs multiple distance metrics as supervised labels for trajectory representation.

- NeuTraj Yao et al. (2019): A RNN-based model that contains spatial attention memory units to model the correlation between trajectories in spatial proximity based on an attention network and an external memory tensor.
- TrajGAT Yao et al. (2022): It uses a GAT-based transformer to capture long term dependencies for GPS trajectory modeling using a deep learning approach that explicitly integrates hierarchical spatial structures and transforms trajectories into graphs for trajectory encoding.
- TrajCL Chang et al. (2023): This framework for trajectory similarity computation leverages contrastive learning, centered on a dual self-attention encoder that integrates structural and spatial features. It utilizes self-supervised pre-training followed by fine-tuning with multi-metric supervision.
- SIMformer Chuang et al. (2024): This method is a single-layer vanilla transformer encoder trained with pairwise MSE loss and equipped with tailored representation similarity functions, to accurately and efficiently approximate free-space trajectory similarities under DTW, Hausdorff, and Fréchet distances.

**Evaluation Metrics.** In the top-$k$ trajectory similarity search task, we evaluate performance using Hit Rate (HR) and Recall (R), following Fang et al. (2022); Yao et al. (2019; 2022). Higher levels of both metrics indicate more accurate results. Given query tokenized trajectories and their ground-truth top-$k$ neighbors based on Euclidean distance, HR measures the proportion of queries retrieving at least one true neighbor within the top-$k$ results (reported as HR@1, HR@5, HR@10), while R measures the fraction of true top-$k$ neighbors retrieved within the top-$k'$ results (reported as R1@5 and R5@10). Note, all retrievals are ranked by the Euclidean distances between learned trajectory representations.

**Implementation Details.** We train Hyper2Edge using the Adam optimizer. The maximum number of training epochs is set to 500, and we early stop training if the HR@5 on the training set did not improve for 10 consecutive epochs. The learning rate was initialized to 0.0001 and reduced by half every 3 epochs upon performance plateaus. In addition, we set the number of spatial tokens $n$ to 10000, the dimension of hidden layer $d$ to 64, the number of top-$k$ to 50, and the temperature coefficients $\tau$ to 0.08 and 0.1 for GeoLife and Porto respectively. The experiments are conducted on a machine with AMD EPYC 7K62 @2.60GHz CPU and one Nvidia A6000 GPU.

## D.2 OVERALL PERFORMANCE (RQ1)

Table 8: Performance of top-$k$ trajectory similarity search on DTW distance.

| Dataset | Method | Ref. | HR@1 | HR@5 | HR@10 | R1@5 | R5@10 |
|---------|--------|------|------|------|-------|------|-------|
| GeoLife | t2vec | ICDE'18 | 13.85% | 18.85% | 23.34% | 27.60% | 28.65% |
| | CL-Tsim | CIKM'22 | 13.96% | 21.96% | 26.36% | 32.29% | 32.81% |
| | HHL-Traj | CIKM'24 | 0.42% | 0.38% | 0.60% | 0.42% | 0.60% |
| | NeuTraj | ICDE'19 | 6.46% | 15.08% | 18.67% | 19.90% | 22.00% |
| | TrajGAT | KDD'21 | 10.10% | 19.10% | 24.83% | 28.23% | 30.83% |
| | TrajCL | ICDE'23 | 2.34% | 12.08% | 18.77% | 23.90% | 28.85% |
| | SIMformer | VLDB'24 | 13.23% | 18.04% | 22.82% | 30.42% | 28.54% |
| | Hyper2Edge | Ours | **18.96%** | **23.73%** | **29.02%** | **37.29%** | **35.94%** |
| Porto | t2vec | ICDE'18 | 5.66% | 6.24% | 6.25% | 11.43% | 8.33% |
| | CL-Tsim | CIKM'22 | 11.02% | 15.20% | 18.02% | 28.31% | 25.83% |
| | HHL-Traj | CIKM'24 | 0.00% | 0.26% | 0.55% | 0.11% | 0.52% |
| | NeuTraj | ICDE'19 | 4.24% | 7.84% | 9.66% | 11.37% | 11.93% |
| | TrajGAT | KDD'21 | 7.64% | 11.05% | 12.76% | 17.14% | 16.74% |
| | TrajCL | ICDE'23 | 0.17% | 13.07% | 17.81% | 18.41% | 25.54% |
| | SIMformer | VLDB'24 | 11.29% | 13.33% | 16.53% | 21.73% | 22.42% |
| | Hyper2Edge | Ours | **13.86%** | **18.36%** | **19.93%** | **30.60%** | **26.27%** |

Table 9: Performance of top-$k$ trajectory similarity search on ERP distance.

| Dataset | Method | Ref. | HR@1 | HR@5 | HR@10 | R1@5 | R5@10 |
|---------|--------|------|------|------|-------|------|-------|
| GeoLife | t2vec | ICDE'18 | 8.75% | 9.79% | 13.91% | 14.90% | 16.56% |
| | CL-Tsim | CIKM'22 | 9.90% | 12.40% | 16.22% | 18.96% | 19.75% |
| | HHL-Traj | CIKM'24 | 0.21% | 0.25% | 0.65% | 0.31% | 0.58% |
| | NeuTraj | ICDE'19 | 1.88% | 5.35% | 7.33% | 7.29% | 8.40% |
| | TrajGAT | KDD'21 | 8.65% | 15.10% | 19.98% | 23.13% | 24.52% |
| | TrajCL | ICDE'23 | 5.37% | 8.96% | 14.96% | 18.42% | 24.48% |
| | SIMformer | VLDB'24 | 8.13% | 9.79% | 13.06% | 15.00% | 15.75% |
| | Hyper2Edge | Ours | **13.75%** | **19.73%** | **22.89%** | **31.77%** | **29.62%** |
| Porto | t2vec | ICDE'18 | 3.90% | 5.15% | 5.21% | 8.26% | 7.06% |
| | CL-Tsim | CIKM'22 | 9.71% | 12.61% | 15.59% | 25.83% | 19.56% |
| | HHL-Traj | CIKM'24 | 0.06% | 0.33% | 0.56% | 0.17% | 0.60% |
| | NeuTraj | ICDE'19 | 2.55% | 4.66% | 5.98% | 6.56% | 7.18% |
| | TrajGAT | KDD'21 | 9.11% | 14.24% | 17.05% | 22.34% | 22.22% |
| | TrajCL | ICDE'23 | 1.34% | 9.56% | 13.59% | 14.70% | 21.66% |
| | SIMformer | VLDB'24 | 7.46% | 13.59% | 15.78% | 23.70% | 19.99% |
| | Hyper2Edge | Ours | **11.48%** | **16.61%** | **17.89%** | **27.55%** | **24.04%** |

## D.3 Cross-Distance Metric Robustness Study (RQ2)

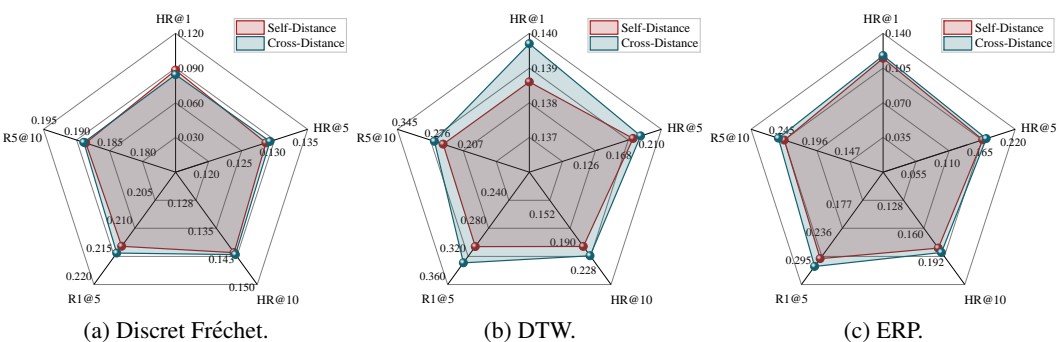

Figure 7: Cross-distance metic robustness on Porto dataset.

## D.4 Ablation Study (RQ4)

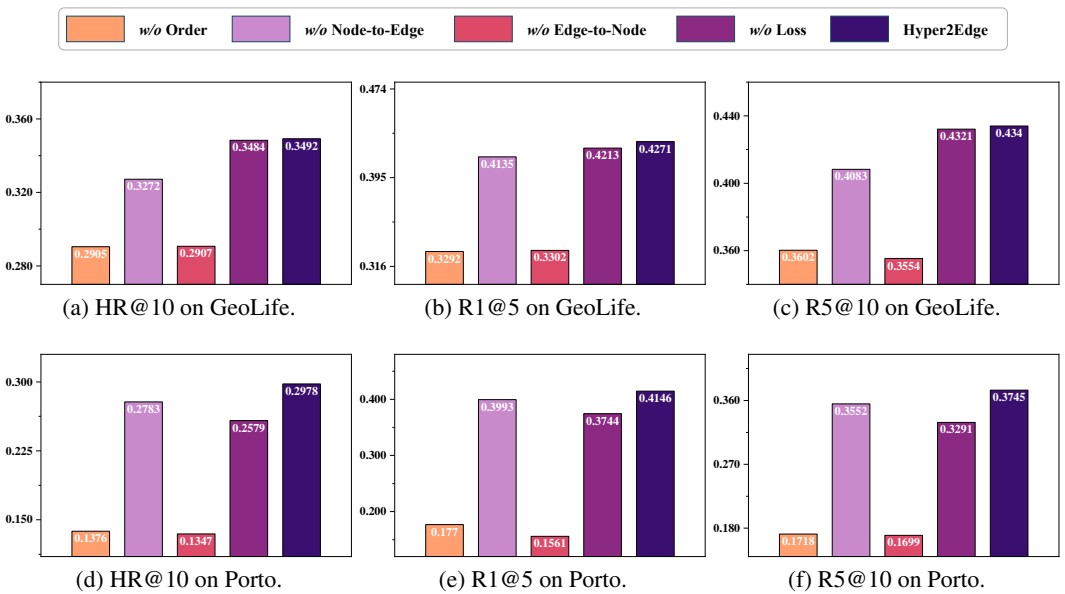

Figure 8: Ablation results by removing components on HR@10, R1@5 and R5@10.

## D.5 Parameter Sensitivity Study (RQ5)

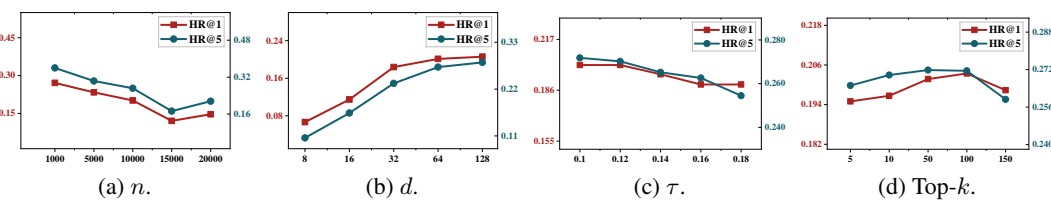

Figure 9: Effect of different $n$, $d$, $\tau$ and Top-$k$ on Porto. The $y$-axis represents hit rate and the $x$-axis is the different hyper-parameter values.

## D.6 GENERALIZATION STUDY IN SPATIO-TEMPORAL CONTEXTS (RQ7)

Table 10: Generalization study in spatio-temporal contexts.

| Dataset | Spatial Distance | Hyper2Edge | HR@1 | HR@5 | HR@10 | R1@5 | R5@10 |
|---|---|---|---|---|---|---|---|
| GeoLife | Euclidean | Only Spatial | 23.54% | 29.67% | 35.53% | 42.92% | 44.25% |
| | | Spatio-temporal | 18.54% | 28.48% | 34.52% | 40.42% | 42.23% |
| | | Difference | **5.00%** | **1.19%** | **1.01%** | **2.50%** | **2.02%** |
| | DTW | Only Spatial | 18.96% | 23.73% | 29.02% | 37.29% | 35.94% |
| | | Spatio-temporal | 13.33% | 23.04% | 28.39% | 34.58% | 34.90% |
| | | Difference | **5.63%** | **0.69%** | **0.64%** | **2.71%** | **1.04%** |
| | ERP | Only Spatial | 13.75% | 19.73% | 22.89% | 31.77% | 29.62% |
| | | Spatio-temporal | 8.96% | 19.06% | 22.61% | 27.50% | 29.25% |
| | | Difference | **4.79%** | **0.67%** | **0.27%** | **4.27%** | **0.38%** |
| Porto | Euclidean | Only Spatial | 20.14% | 27.18% | 29.78% | 41.46% | 37.45% |
| | | Spatio-temporal | 18.67% | 26.20% | 29.12% | 39.14% | 36.57% |
| | | Difference | **1.47%** | **0.98%** | **0.66%** | **2.32%** | **0.88%** |
| | DTW | Only Spatial | 13.86% | 18.36% | 19.93% | 30.60% | 26.27% |
| | | Spatio-temporal | 12.50% | 18.46% | 20.12% | 30.49% | 26.74% |
| | | Difference | **1.36%** | **-0.10%** | **-0.19%** | **0.11%** | **-0.48%** |
| | ERP | Only Spatial | 11.48% | 16.61% | 17.89% | 27.55% | 24.04% |
| | | Spatio-temporal | 11.20% | 16.17% | 17.70% | 27.26% | 23.63% |
| | | Difference | **0.28%** | **0.44%** | **0.19%** | **0.28%** | **0.41%** |

## D.7 SCALABILITY STUDY (RQ8)

Table 11: Scalability study on Porto dataset.

| Trajs Number | HR@1 | HR@5 | HR@10 | R1@5 | R5@10 |
|---|---|---|---|---|---|
| 8839 | 20.14% | 27.18% | 29.78% | 41.46% | 37.45% |
| 20k | 19.35% | 26.74% | 28.71% | 41.45% | 37.18% |

Table 12: Performance of top-$k$ trajectory similarity search on large Porto dataset.

| Method | Ref. | HR@1 | HR@5 | HR@10 | R1@5 | R5@10 |
|---|---|---|---|---|---|---|
| t2vec | ICDE'18 | 5.30% | 5.58% | 5.38% | 10.88% | 7.40% |
| CL-Tsim | CIKM'22 | 13.60% | 17.87% | 19.75% | 30.45% | 26.50% |
| HHL-Traj | CIKM'24 | 0.00% | 0.02% | 0.10% | 0.03% | 0.07% |
| NeuTraj | ICDE'19 | 3.25% | 6.00% | 7.60% | 8.68% | 9.10% |
| TrajGAT | KDD'22 | 14.20% | 19.47% | 21.97% | 31.40% | 29.39% |
| TrajCL | ICDE'23 | OOM | OOM | OOM | OOM | OOM |
| SIMformer | VLDB'24 | 10.03% | 12.53% | 13.81% | 21.63% | 17.62% |
| Hyper2Edge | Ours | **19.35%** | **26.74%** | **28.71%** | **41.45%** | **37.18%** |

