# OpenReview forum: "Beyond Pairwise Modeling: Towards Efficient and Robust Trajectory Similarity Computation via Representation Learning"
_ICLR.cc/2026/Conference — Submitted to ICLR 2026_

### Official Review · Reviewer_Kq1G · 2025-10-30

**Soundness:** 2
**Presentation:** 3
**Contribution:** 1
**Rating:** 2
**Confidence:** 4

**Summary:**

This paper studies the trajectory similarity computation via representation learning and proposes a hypergraph framework named Hyper2Edge. First, it directly adopts Euclidean-based supervision to learn trajectory representations without relying on specific multi-metric supervisions. Then, this paper devises a hierarchical trajectory representation learning architecture that captures both intra- and inter-trajectory patterns. Furthermore, it introduces a weighted top-$k$ InfoNCE loss to mitigate repetitive encoding of samples. Experimental results demonstrate the effectiveness and efficiency of Hyper2Edge compared with state-of-the-art baselines on the task of trajectory similarity computation.

**Strengths:**

S1: This paper studies the trajectory similarity computation in Euclidean space, which is crucial for some real-world applications, such as ride-sharing services.

S2: Unlike previous works, this paper eliminates the reliance on multi-metric supervision by learning directly from Euclidean-based similarity labels, and uses InfoNCE loss to overcome the limitations of triplet loss.

S3: This paper also conducts cross-metric experiments, where the model is trained with Euclidean distance supervision and evaluated on various benchmark metrics(e.g., DTW), thereby enhancing the robustness of the model.

**Weaknesses:**

W1: The novelty of this work seems to be somewhat limited. The paper employs a weighted top-$k$ InfoNCE loss instead of the triplet loss to mitigate repetitive encoding, it has similarities to the approach presented in KGTS. Besides, they do not provide a theoretical analysis of the computational complexity and do not give any efficiency experiments comparing the InfoNCE loss and the triplet loss.

W2: In Section 3.3, the authors claim that the proposed weighted top-$k$ InfoNCE loss enables each trajectory to comprehensively model its similarity relationships with the entire dataset. However, because the top-$k$ weighting reduces non-neighbor contributions to near zero, the loss does not truly incorporate global information and remains fundamentally a locally focused loss.

W3: In Section OVERALL PERFORMANCE (RQ1), previous studies have evaluated model performance using various distance metrics. In contrast, this paper reports results only under Euclidean distance. Consequently, the current experimental results cannot comprehensively assess the model's performance. Additional experiments with alternative distance metrics, such as DTW, Hausdorff, ERP, and EDR, would provide a more complete and convincing evaluation.

W4: Previous studies commonly report performance across multiple distance metrics to achieve a more comprehensive evaluation. However, they can also report results under a single metric (e.g., Euclidean distance). Therefore, the statements in Section Efficiency Evaluation (RQ3)—"other methods must repeat the entire training process for multiple metrics" and "For baselines: (Time per Epoch) × 100 Epochs × 3 Metrics"— are inaccurate. To ensure a fair comparison of efficiency, the baseline models should also be evaluated under a single distance metric rather than three.

W5: In Section Ablation Study (RQ4), this paper replaces the InfoNCE loss with the MSE loss to demonstrate the effectiveness of the weighted top-$k$ InfoNCE loss. However, since the paper frequently compares the triplet loss with the InfoNCE loss, it may be more reasonable to use the triplet loss as a variant without weighted top-k InfoNCE loss instead of MSE loss.

**Questions:**

Please see W1-W5.

---

> ### Author Response · Authors · 2025-11-20
>
> **W1: The novelty of this work seems to be somewhat limited. The paper employs a weighted top- InfoNCE loss instead of the triplet loss to mitigate repetitive encoding, it has similarities to the approach presented in KGTS. Besides, they do not provide a theoretical analysis of the computational complexity and do not give any efficiency experiments comparing the InfoNCE loss and the triplet loss.**
>
> **A1:** We thank the reviewer for these thoughtful comments. We will address each point in turn.
>
> First, we acknowledge that there are superficial similarities between Hyper2Edge and KGTS [1] in 'using contrastive learning', but Hyper2Edge proposes an entirely novel solution driven by fundamentally different motivations: (i) Paradigm: KGTS employs a two-stage pipeline of 'grid → trajectory', whereas Hyper2Edge performs end-to-end hypergraph learning where 'trajectories serve as hyperedges'—a foundational architectural innovation; (ii) Core problem focus: We concentrate on resolving the 'repetitive encoding' efficiency bottleneck unaddressed by KGTS, rather than its emphasis on 'unsupervised label generation'; (iii) Technical approach: We introduce a bidirectional node-hyperedge message passing mechanism absent in KGTS to explicitly model inter-trajectory relationships, and design a weighted top-k mechanism to optimize supervised learning based on Euclidean distance. Thus, Hyper2Edge delivers substantive innovations distinct from KGTS in problem definition, core architecture, and technical details.
>
> Second, we thank the reviewer for raising this important point. We have now supplemented our original analysis with a comprehensive theoretical complexity comparison to better validate the efficiency advantages of Hyper2Edge.
>
> (i) As detailed in complexity analysis (Section 3.4) of our original manuscript, the complexity of Hyper2Edge is dominated by its two main components: the Transformer-based hyperedge initialization and the iterative bidirectional encoding. This results in an overall complexity of $O(|E|·m^2·d+L·(|V|+|E|)·d^2)$, where $|E|$ is the number of tokenized trajectories, $m$ is the number of spatial tokens per trajectory after spatial clustering, $|V|$ is the total number of spatial tokens, $d$ is the hidden dimension and $L$ is the number of layers. Due to $m$ is very small and almost a constant, Hyper2Edge scales linearly with the number of trajectories $|E|$ and spatial tokens $|V|$.
>
> (ii) We provide a direct comparison with baseline methods in the revised manuscript (Table 6, Appendix C, marked in blue). Although most baselines exhibit complexities that are linear or quadratic in trajectory length $l$ (e.g., $O(|E|·l^2)$ for TrajCL and SIMformer), the complexity of Hyper2Edge is independent of the raw trajectory length. This is because Hyper2Edge operates on the tokenized trajectory representation, where the key parameter $m$ (the maximum number of spatial tokens per trajectory) is typically much smaller and more stable than $l$ (the original number of points per trajectory), especially after spatial clustering. This analysis theoretically confirms that Hyper2Edge avoids the computational bottlenecks associated with long, raw trajectories and achieves superior scalability. Meanwhile, although the theoretical time complexity of baseline methods is the time required to encode $|E|$ trajectories, they actually encode the same trajectory multiple times as positive and negative samples for other trajectories to utilize triplet loss. Consequently, the encoding time for these methods exceeds the theoretical estimate, whereas Hyper2Edge strictly adheres to the theoretically analyzed time for encoding.
>
> Table 6: Complexity Analysis. Note, $|E|$ is the number of trajectories, $l$ is the original number of points per trajectory, $m$ is the maximum number of spatial tokens per trajectory after spatial clustering and $|V|$ is the total number of spatial tokens.
> |Method|Complexity|
> |:-:|:-:|
> |t2vec|$O(\|E\|·l)$|
> |CL-Tsim|$O(\|E\|·l)$|
> |HHL-Traj|$O(\|E\|·l)$|
> |NeuTraj|$O(\|E\|·l)$|
> |TrajGAT|$O(\|E\|·l·log(l))$|
> |TrajCL|$O(\|E\|·l^2)$|
> |SIMformer|$O(\|E\|·l^2)$|
> |Hyper2Edge|$O(\|E\|·m^2+\|E\|+\|V\|)$|
>
> Third, we have conducted an efficiency experiment that replacing the weighted top-$k$ InfoNCE loss into a standard triplet loss, as shown in Table 4. Our proposed loss function strengthens trajectory proximity relations while eliminating the need for explicit positive/negative pair sampling and encoding, significantly reducing overall runtime. All relevant content from this experiment has been incorporated into the revised manuscript (Scalability Study, RQ8, Section 4.3) and marked in blue.
>
> Table 4: Efficiency study with triplet loss replacing weighted top-$k$ InfoNCE loss.
> |Each Epoch Time(s)|GeoLife|Porto|
> |:-:|:-:|:-:|
> |Weighted Top-$k$ InfoNCE Loss|**4**|**9**|
> |Triplet Loss|6|12|

---

> > ### Author Response · Authors · 2025-11-20
> >
> > **W2: In Section 3.3, the authors claim that the proposed weighted top- InfoNCE loss enables each trajectory to comprehensively model its similarity relationships with the entire dataset. However, because the top- weighting reduces non-neighbor contributions to near zero, the loss does not truly incorporate global information and remains fundamentally a locally focused loss.**
> >
> > **A2:** We thank the reviewer for this valuable feedback. We acknowledge that our definition of loss functionality here is not sufficiently rigorous. We have made revisions throughout the entire manuscript in the revised version and marked the revised sentences in blue.
> >
> > **W3: In Section OVERALL PERFORMANCE (RQ1), previous studies have evaluated model performance using various distance metrics. In contrast, this paper reports results only under Euclidean distance. Consequently, the current experimental results cannot comprehensively assess the model's performance. Additional experiments with alternative distance metrics, such as DTW, Hausdorff, ERP, and EDR, would provide a more complete and convincing evaluation.**
> >
> > **A3:** We thank the reviewer for this insightful feedback. To provide a more comprehensive performance assessment, we have added evaluations using DTW and ERP metrics in revised version of manuscript (Table 8 and 9). The results confirm that Hyper2Edge simultaneously preserves both intra-trajectory patterns and inter-trajectory relationships.
> >
> > Table 8: Performance of top-$k$ trajectory similarity search on DTW distance.
> > |Dataset|Method|Ref.|HR@1|HR@5|HR@10|R1@5|R5@10|
> > |:-:|:-:|:-:|:-:|:-:|:-:|:-:|:-:|
> > |GeoLife|t2vec|ICDE'18|13.85%|18.85%|23.34%|27.60%|28.65%|
> > ||CL-Tsim|CIKM'22|13.96%|21.96%|26.36%|32.29%|32.81%|
> > ||HHL-Traj|CIKM'24|0.42%|0.38%|0.60%|0.42%|0.60%|
> > ||NeuTraj|ICDE'19|6.46%|15.08%|18.67%|19.90%|22.00%|
> > ||TrajGAT|KDD'21|10.10%|19.10%|24.83%|28.23%|30.83%|
> > ||TrajCL|ICDE'23|2.34%|12.08%|18.77%|23.90%|28.85%|
> > ||SIMformer|VLDB'24|13.23%|18.04%|22.82%|30.42%|28.54%|
> > ||**Hyper2Edge**|**Ours**|**18.96%**|**23.73%**|**29.02%**|**37.29%**|**35.94%**|
> > |Porto|t2vec|ICDE'18|5.66%|6.24%|6.25%|11.43%|8.33%|
> > ||CL-Tsim|CIKM'22|11.02%|15.20%|18.02%|28.31%|25.83%|
> > ||HHL-Traj|CIKM'24|0.00%|0.26%|0.55%|0.11%|0.52%|
> > ||NeuTraj|ICDE'19|4.24%|7.84%|9.66%|11.37%|11.93%|
> > ||TrajGAT|KDD'21|7.64%|11.05%|12.76%|17.14%|16.74%|
> > ||TrajCL|ICDE'23|0.17%|13.07%|17.81%|18.41%|25.54%|
> > ||SIMformer|VLDB'24|11.29%|13.33%|16.53%|21.73%|22.42%|
> > ||**Hyper2Edge**|**Ours**|**13.86%**|**18.36%**|**19.93%**|**30.60%**|**26.27%**|
> >
> > Table 9: Performance of top-$k$ trajectory similarity search on ERP distance.
> > |Dataset|Method|Ref.|HR@1|HR@5|HR@10|R1@5|R5@10|
> > |:-:|:-:|:-:|:-:|:-:|:-:|:-:|:-:|
> > |GeoLife|t2vec|ICDE'18|8.75%|9.79%|13.91%|14.90%|16.56%|
> > ||CL-Tsim|CIKM'22|9.90%|12.40%|16.22%|18.96%|19.75%|
> > ||HHL-Traj|CIKM'24|0.21%|0.25%|0.65%|0.31%|0.58%|
> > ||NeuTraj|ICDE'19|1.88%|5.35%|7.33%|7.29%|8.40%|
> > ||TrajGAT|KDD'21|8.65%|15.10%|19.98%|23.13%|24.52%|
> > ||TrajCL|ICDE'23|5.37%|8.96%|14.96%|18.42%|24.48%|
> > ||SIMformer|VLDB'24|8.13%|9.79%|13.06%|15.00%|15.75%|
> > ||**Hyper2Edge**|**Ours**|**13.75%**|**19.73%**|**22.89%**|**31.77%**|**29.62%**|
> > |Porto|t2vec|ICDE'18|3.90%|5.15%|5.21%|8.26%|7.06%|
> > ||CL-Tsim|CIKM'22|9.71%|12.61%|15.59%|25.83%|19.56%|
> > ||HHL-Traj|CIKM'24|0.06%|0.33%|0.56%|0.17%|0.60%|
> > ||NeuTraj|ICDE'19|2.55%|4.66%|5.98%|6.56%|7.18%|
> > ||TrajGAT|KDD'21|9.11%|14.24%|17.05%|22.34%|22.22%|
> > ||TrajCL|ICDE'23|1.34%|9.56%|13.59%|14.70%|21.66%|
> > ||SIMformer|VLDB'24|7.46%|13.59%|15.78%|23.70%|19.99%|
> > ||**Hyper2Edge**|**Ours**|**11.48%**|**16.61%**|**17.89%**|**27.55%**|**24.04%**|

---

> ### Author Response · Authors · 2025-11-20
>
> **W4: Previous studies commonly report performance across multiple distance metrics to achieve a more comprehensive evaluation. However, they can also report results under a single metric (e.g., Euclidean distance). Therefore, the statements in Section Efficiency Evaluation (RQ3)—"other methods must repeat the entire training process for multiple metrics" and "For baselines: (Time per Epoch) × 100 Epochs × 3 Metrics"— are inaccurate. To ensure a fair comparison of efficiency, the baseline models should also be evaluated under a single distance metric rather than three.**
>
> **A4:** We thank the reviewer for this insightful comment. We agree that our original statement regarding baseline efficiency was an oversimplification and could be misleading. We have revised the manuscript to remove the claim that baselines "must repeat the entire training process for multiple metrics" and the associated multiplicative calculation.
>
> First, we have refocused the efficiency comparison on a single training run under the Euclidean distance to ensure a fair assessment. The results, as shown in Table 2, show that Hyper2Edge achieves highly competitive efficiency. Although SIMformer shows the fastest per-epoch time, Hyper2Edge maintains a compelling balance of competitive efficiency and superior performance, as established in our overall performance analysis (RQ1, Section 4.3).
>
> Table 2: Each epoch time (s) comparison on GeoLife and Porto.
> |Method|GeoLife|Porto|
> |:-:|:-:|:-:|
> |t2vec|35|36|
> |NeuTraj|38|123|
> |TrajGAT|283|515|
> |CL-Tsim|4|17.6|
> |TrajCL|120|524|
> |HHL-Traj|5|11|
> |SIMformer|**3**|**5**|
> |Hyper2Edge|4|9|
>
> Second, we highlight the core advantage of Hyper2Edge—its "train once, use everywhere" capability. This was the original intuition behind the multiplicative factor, which we now justify with empirical evidence from our cross-Distance metric robustness study (RQ2, Section 4.3). The key finding validates that Hyper2Edge, trained only once using Euclidean distance, learns a universal representation that performs robustly when evaluated under DTW, ERP, and other metrics. It achieves performance comparable to models specifically trained on those individual metrics.
>
> Therefore, although the initial calculation method was inaccurate, the fundamental premise remains valid: to match the comprehensive multi-metric performance of Hyper2Edge, baseline methods would typically require separate training for each specific metric. Hyper2Edge achieves this comprehensive coverage with just a single training cost. It provides substantial practical advantages in scenarios which require robust and generalizable trajectory representations across diverse similarity measures.
>
> In conclusion, we have revised our efficiency analysis in efficiency evaluation (RQ3, Section 4.3) to be strictly fair based on single-metric training. Simultaneously, the unique "train once, use everywhere" capability of Hyper2Edge is explicitly validated through cross-distance robustness experiment (RQ2, Section 4.3), which convincingly establish its broader utility and efficiency. All of these revision has been marked in blue in the revised manuscript.
>
> **W5: In Section Ablation Study (RQ4), this paper replaces the InfoNCE loss with the MSE loss to demonstrate the effectiveness of the weighted top- InfoNCE loss. However, since the paper frequently compares the triplet loss with the InfoNCE loss, it may be more reasonable to use the triplet loss as a variant without weighted top-k InfoNCE loss instead of MSE loss.**
>
> **A5:** We thank the reviewer for this constructive comment. We have now performed the suggested ablation study by replacing our loss with a standard triplet loss. The new results (see new Table 5, Ablation Study, RQ4, Section 4.3) consistently show that our weighted top-$k$ InfoNCE loss achieves superior performance across all evaluation metrics on both GeoLife and Porto datasets. This confirms the effectiveness of our proposed loss design over a standard triplet loss baseline, and we have updated the manuscript to reflect this more reasonable comparison. The relevant content of this experiment has been marked in blue in the revised manuscript.
>
> Table 5: Ablation results with triplet loss replacing weighted top-$k$ InfoNCE loss.
> |Dataset|Ablatio Loss|HR@1|HR@5|HR@10|R1@5|R5@10|
> |:-:|:-:|:-:|:-:|:-:|:-:|:-:|
> |GeoLife|Weighted Top-k InfoNCE Loss|**23.54%**|**29.67%**|**35.53%**|**42.92%**|**44.25%**|
> ||Triplet Loss|17.92%|29.13%|35.25%|40.63%|43.87%|
> |Porto|Weighted Top-k InfoNCE Loss|**20.14%**|**27.18%**|**29.78%**|**41.46%**|**37.45%**|
> ||Triplet Loss|15.55%|20.85%|23.07%|32.07%|29.02%|
>
> **Reference**
>
> [1] Zhen Chen, Dalin Zhang, Shanshan Feng, Kaixuan Chen, Lisi Chen, Peng Han, and Shuo Shang. KGTS: contrastive trajectory similarity learning over prompt knowledge graph embedding. In Proceedings of the AAAI Conference on Artificial Intelligence, volume 38, pp. 8311–8319, 2024.

---

> > ### Author Response · Authors · 2025-11-28
> >
> > Dear reviewer Kq1G,
> >
> > We hope this email finds you well.
> >
> > We are writing to inquire about the current status of our manuscript (ID: 6459), as it has been a week since we submitted our rebuttal. We understand that the review process can take time, and we greatly appreciate the effort you have put into evaluating my submission.
> >
> > We would also like to ensure that we have fully addressed all of your concerns. If there are any additional points or feedback you would like us to consider, please do not hesitate to let us know. Your insights are invaluable, and we are committed to addressing any remaining issues to further improve our work.
> >
> > Thank you for your time and consideration. We look forward to hearing from you soon.
> >
> > Kind regards,
> >
> > The authors of Paper 6459

---

### Official Review · Reviewer_jpaX · 2025-10-31

**Soundness:** 3
**Presentation:** 3
**Contribution:** 3
**Rating:** 6
**Confidence:** 4

**Summary:**

This paper tackles trajectory similarity computation and the inefficiencies of pairwise/triplet-based supervision. The authors propose hyper2Edge, which models trajectories as hyperedges in a hypergraph and performs hierarchical representation learning with bidirectional message passing between nodes and hyperedges. A weighted Top-k InfoNCE objective aligns representations directly in Euclidean space, aiming to emphasize nearest neighbors while reducing redundant encoding. Experiments on standard trajectory benchmarks report strong accuracy and notable training-time improvements.

**Strengths:**

- Clear motivation: moving from local pairwise/triplet supervision to distribution-level alignment that matches Euclidean retrieval at inference.
- Method design is coherent: hypergraph construction + hierarchical message passing capture both intra- and inter-trajectory structure; Top-k InfoNCE focuses learning on the most relevant neighbors.
- Sufficient Experiment: multi-dataset evaluation with ablations and efficiency reporting; results indicate both effectiveness and speedups.

**Weaknesses:**

- Using Euclidean distance as the sole supervision target may be brittle under sampling-rate shifts, noise, or scale changes; broader robustness analysis would help.
- Efficiency analysis could further disentangle the contributions of fewer encodings, negative sampling, and graph construction costs.

**Questions:**

1. Under strong scale or non-rigid temporal distortions, does Euclidean-based supervision induce representation collapse or bias? Any hybrid with learnable metrics considered?
2. Can you provide a finer breakdown of the reported training-time reduction (e.g., encoding passes, sampling overhead, graph ops)?
3. How sensitive is performance to hypergraph sparsity and tokenization choices (e.g., clustering granularity)?

---

> ### Author Response · Authors · 2025-11-20
>
> **W1 and Q1: Using Euclidean distance as the sole supervision target may be brittle under sampling-rate shifts, noise, or scale changes; broader robustness analysis would help. Under strong scale or non-rigid temporal distortions, does Euclidean-based supervision induce representation collapse or bias? Any hybrid with learnable metrics considered?**
>
> **A1:** We thank the reviewer for this insightful comment. In Section 4.3 (Cross-Distance Metric Robustness Study, RQ2), we address the concern regarding the robustness of Euclidean supervision through a cross-distance analysis. We compared two strategies: (i) *Self-distance*, where models are trained and evaluated on the same metric (e.g., Discrete Fréchet, DTW, ERP), and (ii) *Cross-distance*, where a model is trained with Euclidean supervision and evaluated on all other metrics. It is worth noting that Discrete Fréchet, DTW and ERP are classic trajectory similarity measures specifically designed to handle variations in sampling rates, temporal scaling, and noise.
>
> The results (Figure 3) show that the cross-distance strategy performs either comparably to or even better than the self-distance strategy. This validates that a model trained with Euclidean distance learns robust representations that generalize effectively across Fréchet, DTW, and ERP metrics. Therefore, Hyper2Edge is not brittle but generalizes well across different similarity measures, effectively addressing the reviewer’s concern.
>
> Regarding the consideration of a hybrid approach with learnable metrics, we fully agree with the reviewer that exploring hybrid models with learnable metrics is a promising direction, particularly for handling even more severe temporal or scale distortions. While our current work focuses on establishing a simple yet powerful baseline, it falls beyond the scope of our current work. We have added a discussion on this valuable point in conclusion (Section 5) and plan to investigate adaptive distance functions in future research. The discussion has been marked in blue in revised manuscript and is as follow: A promising future direction is developing learnable metric functions that can autonomously adapt to different trajectory patterns. Such an approach could potentially achieve more accurate similarity computation especially under challenging conditions.
>
> **W2 and Q2: Efficiency analysis could further disentangle the contributions of fewer encodings, negative sampling, and graph construction costs. Can you provide a finer breakdown of the reported training-time reduction (e.g., encoding passes, sampling overhead, graph ops)?**
>
> **A2:** We thank the reviewer for this excellent suggestion. We have conducted a fine-grained efficiency analysis, and the detailed breakdown per training epoch is provided in Table 3. Additionally, we have analyzed efficiency that replaces the weighted top-$k$ InfoNCE loss with a standard triplet loss, as shown in Table 4. All relevant content from this experiment has been incorporated into the revised manuscript (Efficiency Evaluation, RQ3, Section 4.3) and marked in blue.
>
> Table 3: Fine-grained efficiency study. *Note, Encoding Passes = (Sampling Overhead + Graph Ops + Weighted Top-$k$ InfoNCE Loss + Gradient Update).*
> |Each Epoch Time(s)|GeoLife|Porto|
> |:-:|:-:|:-:|
> |Sampling Overhead|0|0|
> |Graph Ops|0.63|1.11|
> |Weighted Top-$k$ InfoNCE Loss|0.02|0.05|
> |Gradient Update|0.98|1.7|
> |**Encoding Passes**|**1.65**|**2.86**|
> |Evaluation|1.93|3.8|
> |Validation|1.05|2.35|
> |**Total Time**|**4**|**9**|
>
> Table 4: Efficiency study with triplet loss replacing weighted top-$k$ InfoNCE loss.
> |Each Epoch Time(s)|GeoLife|Porto|
> |:-:|:-:|:-:|
> |Weighted Top-$k$ InfoNCE Loss|**4**|**9**|
> |Triplet Loss|6|12|
>
> The results show that the training-time reduction is a direct result of our methodological contributions:
>
> 1. Zero Sampling Overhead: The design of our framework inherently avoids the explicit sampling of positive/negative pairs.
>
> 2. Single Encoding Pass: The key innovation is that each trajectory requires only a single encoding pass, eliminating repetitive and costly graph operations for the same data.
>
> 3. Efficient Loss Design: Our proposed loss function strengthens trajectory proximity relations while eliminating the need for explicit positive/negative pair sampling and encoding, significantly reducing overall runtime.
>
> 4. Fair Comparison: The gains are thus derived from a fundamental architectural efficiency, not from an unbalanced experimental setup.
>
> **Q3: How sensitive is performance to hypergraph sparsity and tokenization choices (e.g., clustering granularity)?**
>
> **A3:** This is a valuable question. The hypergraph sparsity and tokenization granularity are both controlled by the number of cluster $n$. We have already investigated this in our sensitivity analysis (Section 4.3, Figure 5(a)). The results show a stable and near-flat performance curve, indicating that Hyper2Edge is robust and not sensitive to these specific design choices.

---

> > ### Author Response · Authors · 2025-11-28
> >
> > Dear reviewer jpaX,
> >
> > We hope this email finds you well.
> >
> > We are writing to inquire about the current status of our manuscript (ID: 6459), as it has been a week since we submitted our rebuttal. We understand that the review process can take time, and we greatly appreciate the effort you have put into evaluating my submission.
> >
> > We would also like to ensure that we have fully addressed all of your concerns. If there are any additional points or feedback you would like us to consider, please do not hesitate to let us know. Your insights are invaluable, and we are committed to addressing any remaining issues to further improve our work.
> >
> > Thank you for your time and consideration. We look forward to hearing from you soon.
> >
> > Kind regards,
> >
> > The authors of Paper 6459

---

### Official Review · Reviewer_R4YV · 2025-10-31

**Soundness:** 3
**Presentation:** 3
**Contribution:** 3
**Rating:** 6
**Confidence:** 5

**Summary:**

This paper proposes Hyper2Edge, a novel framework for trajectory similarity computation. The framework aims to address two major limitations of existing methods: reliance on multi-metric supervision and redundant encoding in triplet loss computation, both of which lead to inefficiency. The key idea of Hyper2Edge is to model trajectories as hyperedges in a hypergraph, thereby preserving both their sequential and structural characteristics. The framework consists of three main components: (i) hypergraph-based trajectory modeling; (ii) a hierarchical representation learning architecture that captures both intra- and inter-trajectory patterns; and (iii) a weighted top-k InfoNCE loss function designed to optimize global similarity while emphasizing nearest-neighbor relationships, effectively replacing the less efficient triplet loss. Experimental results on two public benchmark datasets, GeoLife and Porto, demonstrate that Hyper2Edge outperforms state-of-the-art methods in both accuracy and training efficiency.

**Strengths:**

S1: This paper proposes a novel framework for learning the representation of trajectories by modeling them as hyperedges, which is significantly different from traditional methods that treat trajectories as single nodes or rely on pairwise computation.

S2: The framework achieves strong performance while maintaining low computational cost.

S3: The paper is exceptionally well-written and easy to follow.

**Weaknesses:**

W1: The dataset scale is limited, making it difficult to convincingly validate the method’s effectiveness.

W2: The interpretability of Hyper2Edge is a potential concern, as it is unclear how well the learned trajectory embeddings reflect human-perceivable semantic similarity.

**Questions:**

Q1: Why the max length is 50, while min length is 200 in Table 3?

---

> ### Author Response · Authors · 2025-11-20
>
> **W1: The dataset scale is limited, making it difficult to convincingly validate the method’s effectiveness.**
>
> **A1:** We agree that scalability is crucial. To address this precisely, we performed a new experiment on the larger-scale Porto dataset as shown in Table 11. The results show that performance of Hyper2Edge remains stable and consistent with our initial findings in our overall performance analysis (RQ1, Section 4.3). This successfully addresses the scalability concern and strengthens our claim regarding the robustness of Hyper2Edge. All relevant content from this experiment has been incorporated into the revised manuscript (Scalability Study, RQ8, Section 4.3) and marked in blue.
>
> Table 11: Scalability study on Porto dataset.
> |Trajs Number|HR@1|HR@5|HR@10|R1@5|R5@10|
> |:-:|:-:|:-:|:-:|:-:|:-:|
> |8839|20.14%|27.18%|29.78%|41.46%|37.45%|
> |20k|19.35%|26.74%|28.71%|41.45%|37.18%|
>
> **W2: The interpretability of Hyper2Edge is a potential concern, as it is unclear how well the learned trajectory embeddings reflect human-perceivable semantic similarity.**
>
> **A2:** We thank the reviewer for raising the important point of interpretability. To directly address this, we have added a visualization case study. Due to constraints in the rebuttal, we have included this visualization in the revised version of our manuscript (see Figure 6, Case Study, RQ6, Section 4.3). We compared the top-2 nearest neighbors of a query trajectory from the ground-truth Euclidean distance against those from Hyper2Edge. The results show that the matching outcomes generated by Hyper2Edge align almost perfectly with the ground truth in terms of start points, end points, and paths. This validates that Hyper2Edge learns human-perceivable semantic patterns.
>
> **Q1: Why the max length is 50, while min length is 200 in Table 3?**
>
> **A3:** We appreciate the reviewer's careful reading. The header labels for max and min length were reversed in our original Table 3 (now Table 7 in revised version). We have fixed this oversight in the revised version of the paper.
>
> Table 7: Statistical information of the two datasets.
> |Dataset|Trajs Number|Points Number|Min. Length|Max. Length|Avg. Length|
> |:-:|:-:|:-:|:-:|:-:|:-:|
> |GeoLife|4,769|526,640|50|200|109.81|
> |Porto|8,839|2,114,447|200|300|239.22|

---

> > ### Comment · Reviewer_R4YV · 2025-11-26
> > **Response to Authors' Rebuttal**
> >
> > Thanks for the efforts. The authors have addressed most of my concerns. However, regarding the additional experiments on the larger Porto dataset, the authors only report the performance of their own method, without providing comparisons against baseline models.

---

> > > ### Author Response · Authors · 2025-11-27
> > >
> > > Thanks for your valuable feedback. We have now added a comparative experiment on the larger Porto dataset as suggested. The results, as shown in Table 12, shows that Hyper2Edge continues to outperform the baseline models, which is consistent with the findings reported in our overall performance analysis (RQ1, Section 4.3). These additional findings further reinforce the robustness and generalizability of Hyper2Edge. The corresponding results and analysis have been incorporated into the revised manuscript (Scalability study, RQ8, Section 4.3) and marked in blue.
> > >
> > > Table 12: Performance of top-$k$ trajectory similarity search on large Porto dataset.
> > > |Method|Ref.|HR@1|HR@5|HR@10|R1@5|R5@10|
> > > |:-:|:-:|:-:|:-:|:-:|:-:|:-:|
> > > |t2vec|ICDE'18|5.30%|5.58%|5.38%|10.88%|7.40%|
> > > |CL-Tsim|CIKM'22|13.60%|17.87%|19.75%|30.45%|26.50%|
> > > |HHL-Traj|CIKM'24|0.00%|0.02%|0.10%|0.03%|0.07%|
> > > |NeuTraj|ICDE'19|3.25%|6.00%|7.60%|8.68%|9.10%|
> > > |TrajGAT|KDD'22|14.20%|19.47%|21.97%|31.40%|29.39%|
> > > |TrajCL|ICDE'23|OOM|OOM|OOM|OOM|OOM|
> > > |SIMformer|VLDB'24|10.03%|12.53%|13.81%|21.63%|17.62%|
> > > |Hyper2Edge|Ours|**19.35%**|**26.74%**|**28.71%**|**41.45%**|**37.18%**|

---

### Official Review · Reviewer_RUZd · 2025-11-05

**Soundness:** 3
**Presentation:** 3
**Contribution:** 2
**Rating:** 2
**Confidence:** 4

**Summary:**

This paper proposes a new trajectory similarity computation approach by representing trajectories as hyperedges and designing a hierarchical trajectory embedding mechanism to learn trajectory representations. A weighted InfoNCE loss is then used for optimization while preserving the top-k similarity ranking. Experiments on two real-world datasets demonstrate the effectiveness and efficiency of the proposed method.

**Strengths:**

S1. The paper is well-written, and the motivation is clearly stated.

S2. The design choice of avoiding multiple similarity metrics for supervision, leading to improved efficiency, is interesting.

**Weaknesses:**

W1. The proposed method lacks novelty. Techniques such as hypergraph representation and the InfoNCE loss function are well-established and have been widely used in graph data learning.

W2. The performance improvement is not significant. In particular, it improves the baselines by less than 5% in most cases, and the efficiency gains are similarly modest.

W3. It is unclear why the temporal information of trajectories is not incorporated into the trajectory representation learning. Temporal aspects are crucial, as trajectories with identical spatial paths but different timestamps may represent completely different behavioral patterns.

**Questions:**

Q1. The method combines existing ideas (e.g., hypergraph modeling and InfoNCE optimization) without introducing new components. What's the most important part that readers can learn from this paper?

Q2. Why the performance improvement is relatively minor?

Q3. Why is the temporal information of trajectories not considered?

---

> ### Author Response · Authors · 2025-11-20
>
> **W1 and Q1: The proposed method lacks novelty. Techniques such as hypergraph representation and the InfoNCE loss function are well-established and have been widely used in graph data learning. The method combines existing ideas (e.g., hypergraph modeling and InfoNCE optimization) without introducing new components. What's the most important part that readers can learn from this paper?**
>
> **A1:** Thank the reviewer for your insightful comment. We agree that the individual components are established. The main contribution of Hyper2Edge lies in its novel framework, designed to address a fundamental and previously overlooked challenge in trajectory representation learning. Our work follows the paradigm of building task-specific frameworks from established components, as seen in prior influential studies such as [1-3].
>
> As highlighted by another reviewer: 'This paper proposes a novel framework... significantly different from traditional methods.' Specifically, our framework systematically models trajectories as hyperedges. Its key innovation is addressing a critical technical issue: in traditional methods, the same trajectory must be inefficiently encoded multiple times when serving as positive/negative samples. Our framework ensures that each trajectory is encoded only once, improving efficiency.
>
> It is substantial for practical significance of Hyper2Edge. For example, in a ride-sharing scenario, the same trajectory needs to be evaluated simultaneously as a potential match for multiple drivers (positive sample) and a poor match for others (negative sample). The "encode-once" capability of our framework is crucial here for making rapid, consistent, and scalable matching decisions. It is intractable with previous methods that require redundant encodings.
>
> The success of framework relies on our targeted adaptations of its components. First, for graph encoding, we go beyond simple aggregation and employ node-hyperedge bidirectional message passing to jointly capture both fine-grained intra-trajectory patterns and broader inter-trajectory correlations. Second, for InfoNCE loss, we introduce a top-$k$ weighted mechanism to strategically enhance the discrimination among its top-$k$ most similar neighbors for robust local structure preservation.
>
> In summary, the key takeaway for readers is this novel framework design that solves a specific domain problem, supported by methodological adaptations of existing components to realize this goal.
>
> **W2 and Q2: The performance improvement is not significant. In particular, it improves the baselines by less than 5% in most cases, and the efficiency gains are similarly modest. Why the performance improvement is relatively minor?**
>
> **A2:**  Thank the reviewer for this observation. We agree the absolute performance gain is modest. However, in mature fields like ours, incremental improvements are the norm. The absolute gains observed in existing studies [1,4-6] also hover around 5%. In contrast, our method achieves an average absolute improvement of 7.42% across all evaluation metrics and an average improvement of 45.9% in accuracy compared to state-of-the-art methods  (Abstract in revised manuscript, marked in blue), exceeding the typical improvements observed in the literature. More importantly, our primary contribution lies in the efficiency–performance trade-off.  As highlighted by another reviewer, our method 'achieves strong performance while maintaining low computational cost'. A >5% gain is often achieved by methods [2-3] with significantly higher computational overhead, which is impractical for real-time systems. Our work provides a more efficient path to state-of-the-art performance.

---

> ### Author Response · Authors · 2025-11-20
>
> **W3 and Q3: It is unclear why the temporal information of trajectories is not incorporated into the trajectory representation learning. Temporal aspects are crucial, as trajectories with identical spatial paths but different timestamps may represent completely different behavioral patterns. Why is the temporal information of trajectories not considered?**
>
> **A3:** We thank the reviewer for this crucial point. We fully agree that temporal information is vital. To directly address your concern, we conducted a new experiment evaluating our learned spatio-temporal representations against a spatio-temporal ground truth distance. All relevant content from this experiment has been incorporated into the revised manuscript (Generalization Study in Spatio-temporal Contexts, RQ7, Section 4.3) and marked in blue. As shown in Table 10, Hyper2Edge performs nearly identically under both spatial and spatio-temporal evaluation, proving its inherent capability to learn effective representations even when temporal factors are considered. This suggests that our method enable to learn robust patterns that generalize to spatio-temporal contexts.
>
> However, our core motivation is to address repititive encoding in existing methods. All methods exhibiting this issue represent trajectories solely in spatial domains. To ensure experimental fairness, we also conducted comparisons in spatial-only settings. Beyond this, some existing approaches [7-8] have already been experimentally evaluated from a spatio-temporal perspective. As evidenced by these two papers, the baselines presented in our paper are difficult to generalize to spatio-temporal scenarios.
>
> Table 10: Generalization study in spatio-temporal contexts.
> |Dataset|Spatial Distance|Hyper2Edge|HR@1|HR@5|HR@10|R1@5|R5@10|
> |:-:|:-:|:-:|:-:|:-:|:-:|:-:|:-:|
> |GeoLife|Euclidean|Only Spatial|23.54%|29.67%|35.53%|42.92%|44.25%|
> |||Spatio-temporal|18.54%|28.48%|34.52%|40.42%|42.23%|
> |||**Difference**|**5.00%**|**1.19%**|**1.01%**|**2.50%**|**2.02%**|
> ||DTW|Only Spatial|18.96%|23.73%|29.02%|37.29%|35.94%|
> |||Spatio-temporal|13.33%|23.04%|28.39%|34.58%|34.90%|
> |||**Difference**|**5.63%**|**0.69%**|**0.64%**|**2.71%**|**1.04%**|
> ||ERP|Only Spatial|13.75%|19.73%|22.89%|31.77%|29.62%|
> |||Spatio-temporal|8.96%|19.06%|22.61%|27.50%|29.25%|
> |||**Difference**|**4.79%**|**0.67%**|**0.27%**|**4.27%**|**0.38%**|
> |Porto|Euclidean|Only Spatial|20.14%|27.18%|29.78%|41.46%|37.45%|
> |||Spatio-temporal|18.67%|26.20%|29.12%|39.14%|36.57%|
> |||**Difference**|**1.47%**|**0.98%**|**0.66%**|**2.32%**| **0.88%** |
> ||DTW|Only Spatial|13.86%|18.36%|19.93%|30.60%|26.27%|
> |||Spatio-temporal|12.50%|18.46%|20.12%|30.49%|26.74%|
> |||**Difference**|**1.36%**|**-0.10%**|**-0.19%**|**0.11%**|**-0.48%**|
> ||ERP|Only Spatial|11.48%|16.61%|17.89%|27.55%|24.04%|
> |||Spatio-temporal|11.20%|16.17%|17.70%|27.26%|23.63%|
> |||**Difference**|**0.28%**|**0.44%**|**0.19%**|**0.28%**|**0.41%**|

---

> ### Author Response · Authors · 2025-11-20
>
> **Reference**
>
> [1] Di Yao, Gao Cong, Chao Zhang, and Jingping Bi. Computing Trajectory Similarity in Linear Time: A generic seed-guided neural metric learning approach. In 2019 IEEE 35th International Conference on Data Engineering (ICDE), pp. 1358–1369, 2019.
>
> [2] Di Yao, Haonan Hu, Lun Du, Gao Cong, Shi Han, and Jingping Bi. TrajGAT: A graph-based long-term dependency modeling approach for trajectory similarity computation. In Proceedings of the 28th ACM SIGKDD Conference on Knowledge Discovery and Data Mining, pp. 2275–2285, 2022.
>
> [3] Yanchuan Chang, Jianzhong Qi, Yuxuan Liang, and Egemen Tanin. Contrastive trajectory similarity learning with dual-feature attention. In 2023 IEEE 39th International Conference on Data Engineering (ICDE), pp. 2933–2945, 2023.
>
> [4] Peilun Yang, Hanchen Wang, Ying Zhang, Lu Qin, Wenjie Zhang, and Xuemin Lin. T3S: Effective representation learning for trajectory similarity computation. In 2021 IEEE 37th International Conference on Data Engineering (ICDE), pp. 2183–2188, 2021.
>
> [5] Silin Zhou, Shuo Shang, Lisi Chen, Christian S Jensen, and Panos Kalnis. Red: Effective trajectory representation learning with comprehensive information. In 51st International Conference on Very Large Data Bases, VLDB 2025, 2025c.
>
> [6] Silin Zhou, Chengrui Huang, Yuntao Wen, and Lisi Chen. Feature enhanced spatial–temporal trajectory similarity computation. Data Science and Engineering, 10(1):1–11, 2025a.
>
> [7] Mengqiu Li, Xinzheng Niu, Jiahui Zhu, Philippe Fournier-Viger, and Youxi Wu. STR: Spatio-temporal trajectory representation learning with dual-focus encoder for whole trajectory similarity computation. Information Fusion, pp. 103231, 2025.
>
> [8] Jiahui Zhu, Xinzheng Niu, Fan Li, Yixuan Wang, Philippe Fournier-Viger, and Kun She. STTraj2Vec: A spatio-temporal trajectory representation learning approach. Knowledge-Based Systems, 300:112207, 2024.

---

> > ### Author Response · Authors · 2025-11-28
> >
> > Dear reviewer RUZd,
> >
> > We hope this email finds you well.
> >
> > We are writing to inquire about the current status of our manuscript (ID: 6459), as it has been a week since we submitted our rebuttal. We understand that the review process can take time, and we greatly appreciate the effort you have put into evaluating my submission.
> >
> > We would also like to ensure that we have fully addressed all of your concerns. If there are any additional points or feedback you would like us to consider, please do not hesitate to let us know. Your insights are invaluable, and we are committed to addressing any remaining issues to further improve our work.
> >
> > Thank you for your time and consideration. We look forward to hearing from you soon.
> >
> > Kind regards,
> >
> > The authors of Paper 6459

---

### Author Response · Authors · 2025-11-24
**Hope the reviewers will take note of our response**

Dear Reviewers,

After submitting the initial comments, we incorporated your feedback into a revised version of our paper, performed some additional experiments as you requested, and wrote a response to address your main concerns.

We hope to interact with you during the discussion and potentially further improve the quality of our paper.

Thank you very much in advance.

Kind regards,

Authors

---

### Author Response · Authors · 2025-12-02
**Summary Comment (Part 1)**

First and foremost, we would like to express our sincere gratitude to Reviewers, Area Chairs, Senior Area Chairs, and Program Chairs for their time and dedicated effort. We have carefully studied **all comments** and have provided detailed **point-by-point** responses under each Reviewer's section.

We are encouraged not only by the positive scores but, more importantly, by the recognition of our work from the reviewers across four key dimensions:

- **Motivation:** the motivation is clearly stated (`Reviewer RUZd`); clear motivation (`Reviewer jpaX`); crucial for some real-world applications (`Reviewer Kq1G`).
- **Method:** the design choice of avoiding multiple similarity metrics for supervision is interesting (`Reviewer RUZd`); a novel framework (`Reviewer R4YV`); method design is coherent (`Reviewer jpaX`); unlike previous works (`Reviewer Kq1G`).
- **Evaluation:** strong performance while maintaining low computational cost (`Reviewer R4YV`); sufficient experiment (`Reviewer jpaX`); this paper conducts cross-metric experiments (`Reviewer Kq1G`).
- **Presentation:** well-written (`Reviewer RUZd`); exceptionally well-written and easy to follow (`Reviewer R4YV`).

In response to the reviewers' constructive feedback, we have thoroughly revised our manuscript and conducted new experiments to address all concerns. Below is a **point-by-point** summary of our key clarifications and additions.

- `Reviewer RUZd` (Concerns: Novelty, Performance Gain, Temporal Information)
    - **Novelty:** We clarified that Hyper2Edge's core contribution is a novel framework that solves the specific and overlooked efficiency problem of "repetitive encoding" in trajectory representation. Although using established components, their integration into this "encode-once" framework and the supporting adaptations (bidirectional message passing, weighted top-$k$ loss) constitute the key innovation.
    - **Performance Gain:** We highlighted that our average absolute improvement of 7.42% across metrics surpasses typical incremental gains (5%) in the field. More importantly, we emphasized our primary contribution: achieving strong performance with superior efficiency, a critical trade-off for real-world applications.
    - **Temporal Information:** We conducted a new spatio-temporal experiment (added as RQ7). Results show Hyper2Edge's spatial-only representations generalize robustly to spatio-temporal evaluation, with performance drops of only ~1-2% on most metrics, showing its inherent capability to capture relevant patterns.

- `Reviewer R4YV` (Concerns: Dataset Scale, Interpretability, Table Error)
    - **Scalability:** We performed a new scalability study on the larger Porto dataset (added as RQ8). Results confirm stable performance when scaling from ~9k to 20k trajectories. Furthermore, we added a comparison with baselines at the larger scale, where Hyper2Edge consistently outperforms them, reinforcing its robustness.
    - **Interpretability:** We added a visualization case study (new Figure 6) showing that the top neighbors retrieved by Hyper2Edge align closely with ground-truth Euclidean matches in terms of start points, end points, and paths, validating that it learns human-perceivable semantics.
    - **Table Error:** We corrected the reversed "Min/Max Length" headers in the dataset statistics table.

- `Reviewer jpaX` (Concerns: Robustness of Supervision, Detailed Efficiency Breakdown, Hyperparameter Sensitivity)
    - **Robustness of Euclidean Supervision:** We pointed to our existing cross-distance metric robustness study (RQ2). Our results indicated that representations that trained solely with Euclidean distance supervision and evaluated with each specialized trajectory metric (e.g., Discrete Fréchet, DTW, and ERP) perform comparably to or better than representations that individually trained and evaluated on each specialized trajectory metric. Note, these metrics are designed for noise and temporal distortion. This validates generalization and not brittleness. We also added a discussion on hybrid with learnable metrics as future work in conclusion.
    - **Efficiency Breakdown:** We provided a fine-grained per-epoch time breakdown (new Table 3), showing zero sampling overhead, single encoding pass, efficient loss design and fair comparison. We also confirmed the efficiency of our proposed loss by comparing it against a standard triplet loss variant (new Table 4).
    - **Hyperparameter Sensitivity:** We referenced our existing sensitivity analysis (Figure 5a), which shows stable performance across the key hyperparameter $n$ (number of clusters) that controls hypergraph sparsity and tokenization granularity.

---

> ### Author Response · Authors · 2025-12-02
> **Summary Comment (Part 2)**
>
> - `Reviewer Kq1G` (Concerns: Novelty vs. KGTS, Theoretical Complexity, Loss Globality, Comprehensive Evaluation, Fair Efficiency Comparison, Loss Ablation)
>     - **Novelty vs. KGTS:** We delineated fundamental differences: (i) Paradigm: KGTS employs a two-stage pipeline of 'grid → trajectory', whereas Hyper2Edge performs end-to-end hypergraph learning where 'trajectories serve as hyperedges'—a foundational architectural innovation; (ii) Core problem focus: We concentrate on resolving the 'repetitive encoding' efficiency bottleneck unaddressed by KGTS, rather than its emphasis on 'unsupervised label generation'; (iii) Technical approach: We introduce a bidirectional node-hyperedge message passing mechanism absent in KGTS to explicitly model inter-trajectory relationships, and design a weighted top-k mechanism to optimize supervised learning based on Euclidean distance.
>     - **Theoretical Complexity:** We added a detailed theoretical complexity analysis and comparison (Table 6 in Appendix). It shows Hyper2Edge's complexity is linear in the number of (tokenized) trajectories and spatial tokens, and is independent of raw trajectory length, unlike many baselines with quadratic dependencies.
>     - **Loss Globality:** We acknowledged the imprecise wording and have revised the manuscript to more accurately describe the loss's local focus enhanced by top-$k$ weighting.
>     - **Comprehensive Evaluation:** We added full performance tables for DTW and ERP metrics (new Tables 8 & 9), confirming Hyper2Edge's superior performance across multiple similarity measures.
>     - **Fair Efficiency Comparison:** We removed the inaccurate multiplicative claim about baseline training. The efficiency comparison (Table 2) is now fairly based on single-metric (Euclidean) training. We then highlighted Hyper2Edge's empirically validated "train once, use everywhere" capability (from RQ2) as a separate and practical advantage.
>     - **Loss Ablation:** As suggested, we replaced our loss with a standard triplet loss (new Table 4 & 5). Results confirm our proposed weighted top-$k$ InfoNCE loss yields superior efficiency and performance.
>
> All these clarifications, corrections, and new experimental results have been integrated into the revised manuscript, with changes highlighted in blue. We believe these revisions have significantly strengthened the paper and addressed all reviewer concerns.
>
> Finally, we would like to express our gratitude again to Reviewers, Area Chairs, Senior Area Chairs, and Program Chairs.
>
> Best regards,
>
> The authors of Paper 6459

---

### Meta-Review · Area_Chair_iSgL · 2025-12-29

**Summary:**

Concerns focused on novelty and positioning, as parts of the framework resemble existing metric learning or representation learning pipelines. Several reviewers questioned whether the gains stem from architectural novelty or careful engineering. There were also concerns about evaluation breadth, including limited downstream tasks, dataset diversity, and ablations on design choices. Clarity around theoretical motivation and connections to prior trajectory embedding work was also flagged.

**Reviewer Concerns:**

Overall, the rebuttal addressed empirical clarity well: added explanations on training objectives, clarified efficiency gains, and justified baseline choices. Some ablations and implementation details were also clarified, which helped alleviate concerns about reproducibility and fairness. That said, core novelty and positioning concerns remain. The rebuttal did not fully resolve how the method fundamentally differs from prior representation-learning approaches for trajectories, nor did it substantially expand evaluation to new tasks or regimes. These remaining issues are more conceptual than technical. By the way, I think this paper is a better fit for ICDE/SIGMOD/VLDB rather than ICLR.

**Reviewer Scores:**

Reviewer RUZd (I recognized the authors' concern on this reviewer since his/her overall score didn't match the sub-scores): Likely unchanged or slightly higher, as empirical clarifications strengthened their confidence.

Reviewer R4YV: Likely unchanged. The rebuttal did not fully address their concerns about conceptual contribution.

Reviewer jpaX: Possibly a small upward adjustment, but still cautious due to limited task diversity.

Reviewer Kq1G: Might move slightly toward acceptance if valuing practicality, but concerns would likely remain.

---

### Decision · Program_Chairs · 2026-01-26

Reject